# A Review of Biologically Active Oxime Ethers

**DOI:** 10.3390/molecules28135041

**Published:** 2023-06-28

**Authors:** Tomasz Kosmalski, Daria Kupczyk, Szymon Baumgart, Renata Paprocka, Renata Studzińska

**Affiliations:** 1Department of Organic Chemistry, Faculty of Pharmacy, Collegium Medicum in Bydgoszcz, Nicolaus Copernicus University in Toruń, Jurasza Str. 2, 85-089 Bydgoszcz, Poland; sz.baumgart@cm.umk.pl (S.B.); renata.bursa@cm.umk.pl (R.P.); 2Department of Medical Biology and Biochemistry, Faculty of Medicine, Collegium Medicum in Bydgoszcz, Nicolaus Copernicus University in Toruń, Karłowicza Str. 24, 85-092 Bydgoszcz, Poland; dariak@cm.umk.pl

**Keywords:** oxime ethers, antimicrobial activity, antidepressive activity, anticancer activity, siponimod, fluvoxamine, ridogrel, oxiconazole

## Abstract

Oxime ethers are a class of compounds containing the >C=N-O-R moiety. The presence of this moiety affects the biological activity of the compounds. In this review, the structures of oxime ethers with specific biological activity have been collected and presented, and bactericidal, fungicidal, antidepressant, anticancer and herbicidal activities, among others, are described. The review includes both those substances that are currently used as drugs (e.g., fluvoxamine, mayzent, ridogrel, oxiconazole), as well as non-drug structures for which various biological activity studies have been conducted. To the best of our knowledge, this is the first review of the biological activity of compounds containing such a moiety. The authors hope that this review will inspire scientists to take a greater interest in this group of compounds, as it constitutes an interesting research area.

## 1. Introduction 

Oxime ethers are a class of compounds containing the >C=N-O-R moiety. They are formed in the reaction of oxime (ketones or aldehydes derivatives) with alkyl or aryl halogen. The formation of an oxime ether moiety is one way to combine different structural elements into one molecule. This gives the possibility of composing compounds that exhibit various biological activity. The literature presents many oxime ether compounds with diverse biological activity. The characteristic ether moiety of the oxime is a structural element in the molecules of known drugs. An example is oxiconazole (**1**)—an antifungal drug in the form of a salt (nitrate(V)) (Figure 1). It is an acetophenone-oxime derivative of the basic imidazole structural unit (substituted heterocyclic ring with a nitrogen in the 3-position). It is applied to the skin in the form of an ointment, cream or powder [1]. Fluvoxamine (**2**) is an antidepressant which functions pharmacologically as a selective serotonin reuptake inhibitor. It is a derivative of (*E*)-5-methoxy-1-(4-(trifluoromethyl)phenyl)pentan-1-one oxime recently also used in the treatment of COVID-19 [2]. Ridogrel (**3**)—an oxime ether derivative of (*E*)-pyridin-3-yl(3-(trifluoromethyl)phenyl)methanone—is an orally active, potent and specific combined thromboxane synthase inhibitor and thromboxane A2 receptor (thromboxane/prostaglandin endoperoxide receptor) antagonist. It is a dual-action drug useful for the prevention of systemic thrombo-embolism and as an adjunctive agent to thrombolytic therapy in acute myocardial infarction [3]. Siponimod (**4**), a benzyl ether derivative of (*E*)-1-(2-ethyl-4-(1-(hydroxyimino)ethyl)benzyl)azetidine-3-carboxylic acid, Mayzent^®^ is a sphingosine 1-phosphate receptor modulator indicated for the treatment of relapsing forms of multiple sclerosis (MS), to include clinically isolated syndrome, relapsing-remitting disease and active secondary progressive disease, in adults. It works by reducing the number of lymphocytes in the peripheral blood to minimize the migration of lymphocytes into the central nervous system [4]. The presence of the oxime ether group in drugs used in various diseases is an incentive to search for new compounds containing this structural fragment and to study their various biological activities. The oxime ethers have been tested, among others, in order to assess their antifungal, antibacterial, antidepressant, anti-inflammatory, anticancer and other activities. This manuscript is a review of the activity of various oxime ethers. To the best of our knowledge, this is the first listing of the activity of compounds containing such a moiety. The subject matter of this *review* is important, interesting and necessary. It fills a certain thematic gap devoted to the biological activities of the oxime ethers. At the same time, it expresses research interest in this class of compounds in a specific way, and it also indicates the scientific potential of the presented chemical structures and their biological activities. 

## 2. Oxime Ethers with an Antifungal Activity

Oxiconazole nitrate (**1**, Figure 1) (under the brand name Oxistat), a derivative of imidazole, is an oxime ether with antifungal properties that is well known and widely used in medicine. Oxiconazole is an antifungal drug for the treatment of skin infections, marketed worldwide. It is active against dermatophytes and yeasts, e.g., *Trichophyton rubrum*, *Trichophyton mentagrophytes*, *Epidermophyton floccosum* and *Malassezia furfur*. It is used topically in the treatment of superficial mycosis (including athlete’s foot, skin) in the form of a cream or lotion [5,6]. 

Rossello et al. [7] studied oxime ethers of imidazole derivatives (**5**) and (**6**) (Figure 2), which differed in the distribution of substituents around the C=N-O group compared to oxiconazole (**1**). They were evaluated in terms of antifungal properties, e.g., against the most important human pathogens—*Candida albicans* and *Aspergillus fumigatus*, as well as *Candida glabrata*, *Candida krusei* and *Aspergillus flavus*. For this purpose, MIC (minimum inhibitory concentrations) and MFC (minimum fungicidal concentrations) values of the tested compounds were determined, and their effects were compared with fluconazole and oxyconazole. Against *C. albicans* and *A. fumigatus*, derivatives **5a** and **5b** were the most active, with MIC values in the range of 2–8 µg/mL. In the case of the remaining fungal strains, the strongest activity against *C. glabrata* and *Candida parapsilosis* was found in compound **5c** (MIC: 0.06 µg/mL and 0.004 µg/mL, respectively), and against *A. flavus*, *T. mentagrophytes* and *T. rubrum*, compound **5b** (MIC values: 8, 4 and 2 µg/mL, respectively). The authors showed that the most promising results were obtained by substituting the *N-*ethoxy-morpholino group in the R position (at the carbon atom of the oxime ether moiety). However, the introduction of fluorine atoms into the phenyl rings and the introduction of the *N*-methyleneimidazole group at the carbon atom of the oxime ether group turned out to be unfavorable. Such an oxiconazole derivative (**6**) was significantly weaker (MIC for *C. albicans*, *A. fumigatus* and *A. flavus*: 2, >128 and >128 μg/mL, respectively). Among the tested compounds, the highest activity was shown by compound **5c** against *C. parapsilosis*. Only in this case was a higher MIC than the reference compound, oxiconazole, observed. The remaining compounds showed lower activity against the tested species of fungi than the reference pacompounds [7]. 

Emami et al. [8] obtained and tested in vitro the antifungal activity of the (*E*) and (*Z*) stereoisomers of the oxime ethers 1,2,4-(**7a**,**b**) and 1,3,4-triazolylchromanone (**7c**) (Figure 2). The compounds were tested for antifungal activity against *C. albicans*, *Saccharomyces cerevisiae*, *Aspergillus niger* and *Microsporum gypseum*, using fluconazole and oxiconazole as reference drugs. Taking into account the structure–activity relationship, the study revealed that the 1,2,4-triazol-1-yl chromanone oxime ether derivatives (**7a**,**b**) performed better than the 1,3,4-triazol-4-yl chromanone derivatives (**7c**). Many compounds showed antifungal activity comparable to or higher than fluconazole and oxiconazole against the fungal strains used in the study. The most potent against *C. albicans* and *S. cerevisiae* was the (*Z*) isomer of compound **7a**, with an MIC of 4 and 8 μg/mL, respectively, and against *A.niger*, the most potent was the stereoisomer (*E*)-**7b** (MIC = 1 μg/mL). Against *M. gypseum*, the best MIC value, reaching 1 μg/mL, was recorded for both geometric isomers of compound **7b**. Analyzing the results for 1,3,4-triazol-4-yl derivatives (**8**), promising data were collected for (*Z*)-**7c** (MIC values in the range of 16–64 μg/mL). There was no significant difference in the activity between the (*E*) and (*Z*) stereoisomers of the tested compounds against strains of *C. albicans*, *S. cerevisiae* and *M. gypseum*; only the MIC values for *A. niger* showed divergences in the activity of most (*E*)- and (*Z*)-isomers, but it was noted that the type, number and position of the halogen substitution on the *O-*benzyl group and the presence of a chlorine atom in the 7-position of the chroman ring have a differentiated effect on the antifungal activity of the tested oxime ethers [8]. 

Emami et al. also tested the in vitro antifungal activity of (*Z*)-trans-3-azolyl-2-methylchromanone oxime ethers (**8**) [9] (Figure 2). The authors suggested that the methyl group in position 2 of the chroman ring contributes to the better antifungal activity of these compounds, and its effect on the activity also depends on other substituents. Among the oxime ethers analyzed in the study, the best MIC values were obtained by compound **8a**. They were, respectively: 0.25 μg/mL for *C. albicans*, 0.5 μg/mL for *S. cerevisiae*, 8 μg/mL for *A. niger* and 1 μg/mL for *M. gypseum*. The derivative **8a** turned out to be more active than the reference drugs for all fungal species tested, except for *A. niger*. The derivative **8b** containing an additional chlorine atom (R substituent) showed slightly lower activity against *C. albicans* and *S. cerevisiae* (MIC 0.5 and 2 μg/mL, respectively) compared to compound **8a**. The authors also showed that the analogous triazole derivatives were less-active molecules compared to the imidazole derivatives [9]. The oxime ethers, imidazolylchromanone derivatives [10] with a phenoxyethyl group, show high activity against the *Cryptococcus gatti* strain. Compound **9** is characterized by the highest activity among the six tested derivatives, equal to 0.5 μg/mL. It also exhibits antifungal activity against *A. fumigatus* (MIC = 32 μg/mL). For comparison, the MIC values for the reference drug Itraconazole are 0.5 and 4 μg/mL, respectively [10]. 

Demirayak et al. [11] studied the antifungal properties of aryl(benzofuran-2-yl)ketoxime ethers (**10**) against *C. albicans*. The obtained results allow us to conclude that these ethers are characterized by a satisfactory antifungal effect, often comparable to known antifungal drugs. The most active turned out to be compounds **10a** and **10b**, for which the MIC is 1 µg/mL, which is also the value obtained for oxiconazole and fluconazole. 

Xu et al. received a series of new *O*-benzyl ethers of azole oximes piperidin-4-one derivatives (**11**) showing very high antifungal activity against the following strains: *C. albicans*, *Candida tropicalis*, *C. parapsilosis*, *C. krusei*, *Cryptococcus neoformans*, *T. rubrum* and *A. fumigatus* [12]. Among the tested derivatives, compound **11a** shows the highest activity against *C. albicans*, *C. tropicalis*, *C. parapsilosis* and *C. krusei* (MIC_80_ values are: 0.004, 0.001, 0.004 and 0.0625 μg/mL, respectively). In the first three cases, it is a value comparable to or lower than that obtained for the reference compound—Voriconazole. On the other hand, compound **11b**, containing Cl in the 3-position of the phenyl ring, is the most active against *C. neoformans* and *T. rubrum*. The MIC_80_ values are 0.001 and 0.016 μg/mL, respectively. An interesting compound is also **11c**, containing a fluorine atom at C-2 of the phenyl ring. This compound shows high activity against *C. tropicalis* and *C. neoformans*, comparable to or higher than the reference drug [12]. 

Parthiban et al. [13] tested the activity against *C. albicans*, *C. parapsilosis*, *A. niger* and *C. neoformans* of 25 *O*-methyloximes, derivatives of substituted 2,4,6,8-tetraaryl-3,7-diazabicyclo[3.3.1]nonan-9-one (**12**, Figure 2). Two compounds showed particular activity: one containing a fluorine atom in position 4 and one a chlorine atom in position 2. These compounds were more active than Fluconazole against *A. niger* (MIC_90_ = 2 µg/mL vs. 4 µg/mL for Fluconazole). The same authors also studied the antifungal activity of *O*-benzyl ethers of oximes, derivatives of piperidin-4-one, tetrahydropyran-4-one and tetrahydrothiopyran-4-one. Among the 28 compounds, the most active was the piperidine derivative (**13a**), with a methoxy group in the position of 4 phenyl rings, which, in the case of most of the tested strains (*C. albicans*, *Candida-51*, *Rhizopus* sp., *A. niger*, *A. flavus* and *C. neoformans*), was characterized by lower MIC values than Amphotericin B [14]. The introduction of a methyl group to the nitrogen atom of the piperidine ring did not improve antifungal properties. Only the derivative (**13b**) containing a bromine atom in position 2 of the benzyl substituent showed better activity against *A. flavus* than Amphotericin B [15]. 

1-(2-Naphthyl)-2-(imidazole-1-yl)ethanone oxime ethers (**14**) (Figure 2) were also tested for their antifungal activity against *C. albicans*, *C. krusei* and *C. parapsilosis*. The *E*-isomer of the ethyl ether turned out to be the most active, for which the MIC values were, respectively, 1, 1 and 2 μg/mL, and in the case of *C. krusei*, these values turned out to be lower than for Fluconazole [16].

A summary of the antifungal activity of the selected oxime ethers is presented in Figure 3 and in Appendix A. The largest group of compounds was tested against *C. albicans*. Compound **8a** has the lowest MIC value. (Figure 3). However, among the described group of the oxime ethers, compounds **11a**–**11c** showed the strongest antifungal activity, of which compound **11a** (expressed as MIC_80_) was the most active (Appendix A). 

The oxime ethers were also tested for their use as fungicides in plant protection. Pyribencarb (**15**) is an oxime ether and is a fungicide used against a wide range of plant pathogenic fungi, especially gray mold caused by *Botrtis cinerea* and *Sclerotinia sclerotirum* stem rot (Figure 4) [17]. 

Cymoxanil (Curzate, **16**, Figure 4) is selective against the certain fungi belonging to the *Peronosporales*. It is effective for the protectant and postinfection control of *Phytophora infestans* on potato and tomato and *Plasmora viticola* on grapevine [18]. 

Xie et al. [19] bioassayed 21 new oxime ethers—enoxastrobin derivatives containing an indole group (**17**). Enoxastrobin is a fungicide used for control of leaf blotch, leaf rust and powdery mildew on wheat and other fungal diseases on cucumbers, tomatoes and grapes [20]. The following fungal strains were used in the study: *Pyricularia oryzae*, *B. cinerea*, *Erysiphe graminis*, *Colletotrichum lagenarium*, *Pseudoperonospora cubensis* and *Puccinia sorghi Schw*. In vitro studies were performed with *P. oryzae* and *B. cinerea*, using enoxastrobin derivatives at a concentration of 25 mg/L. In the case of the remaining species, in vivo studies were performed using the tested compounds at a concentration of 100 mg/L. All derivatives showed different fungicidal activity, with the best activity observed against *P. oryzae*, *E. graminis*, *C. lagenarium* and *P. sorghi Schw*. The activity of most of the tested compounds against these fungal species was higher than the activity of the reference compound (Enoxastrobin). Particularly noteworthy is the derivative containing a chlorine atom in the phenyl ring and the substituent Q1 (methyl 3-methoxy-2-methylacrylate), which also shows high (over two times higher than the reference compound) activity against *P. cubensis* [19]. 

Analogous enoxastrobin derivatives containing a benzothiophene group were tested for antifungal activity by the same group [21]. All tested compounds are characterized by high activity against *Pyricularia oryzae* (% inhibition ≥ 50% at a concentration of 6.25 mg/L) and *B. cinerea* (100% inhibition at a concentration of 6.25 mg/L), comparable to or higher than that of enoxastrobin. An in vivo fungicidal test at 100 mg/L showed that some ethers performed better against *E. graminis*, *C. lagenarium* and *P. sorghi Schw* but worse against *P. cubensis*, compared to enoxastrobin. The ability to inhibit the growth of *E. graminis*, *C. lagenarium* and *P. sorghi Schw*. was also tested at lower concentrations (0.39–6.25 mg/L), and the obtained results revealed a satisfactory effect, e.g., of methyl 3-methoxypropenoate derivatives on given fungal strains, mainly at a concentration of 1.56 mg/L. Compound **18** (Figure 4), substituted with 6-chlorobenzothiophene, had the most promising fungicidal properties, at a concentration of 0.39 mg/L, and it inhibited most of the tested fungi more strongly than enoxastrobin [21]. 

Zhang et al. [22] obtained a new series of derivatives of acetophenone and benzaldehyde oxime ethers, and then they tested their activity against *Rhizoctonia solani* and *S. sclerotiorum*. Some of the analyzed oxime ethers at a concentration of 50 µmol/L showed activity against *R. solani*. The most active was *O*-4-bromobenzyl ether of the 2,3-difluorobenzaldehyde oxime (**19**) (EC_50_ = 8.5 µg/mL vs. 7.3 µg/mL for the reference compound Thiophanate-methyl). However, no promising activity of the tested derivatives against *S. sclerotiorum* was observed in the concentration range up to 100 µg/mL. 

Hu et al. [23] synthesized *O*-benzyl oxime ethers containing *β*-methoxyacrylate moiety with high fungicidal activity against *E. graminis*. The most active derivative **20** showed 100% activity at concentrations up to 3.15 mg/L, while the reference compound—azoxystrobin—showed only 70% activity at this concentration. 

In the patent of Wenderoth et al. [24], oxime ethers with fungicidal activity were developed. An exemplary study of the fungicidal activity was carried out on leaves of seedlings of “Jubilar” wheat infected with *Septoria nodorum* spores, and it was observed that high activity (90%) was shown by derivative **21** at a concentration of 0.05%. (Figure 4).

Sun et al. [25] obtained a series of new triazole derivatives containing an oxime ether group—compounds with a broad spectrum of antifungal activity. An analysis of the (*E*) and (*Z*) isomers of the new products indicated that the (*Z*) isomers had higher activity compared to the (*E*) isomers. Compound (*Z*)-**22** showed very high in vitro activity against *R. solani* (EC_50_ = 0.41 μg/mL, vs. EC_50_ = 0.55 μg/mL for carbendazim) and preventive efficacy (94.58% at a concentration of 200 μg/mL) (Figure 4).

Li et al. tested the antifungal activity of (*Z*)-3-carene oxime ethers derivatives against eight fungal strains, e.g., *R. solani*, *Physalospora piricola* and *Cercospora arachidicola*. Some derivatives showed promising antifungal activities. Particularly noteworthy are the derivatives **23a** and **23b** (Figure 4), which were characterized, respectively, by 98.4% (R = 2-CH_3_) and 94.9% (R = 3-Cl) inhibition against *R. solani* (at a concentration of 50 μg/mL), comparable to chlorothalonil [26].

Adamantane derivatives of oxime ethers with a 3-pyridyl substituent **24a-c**, developed by Liu et al. [27], are compounds with the high antifungal activity. Compounds **24a** and **24b** (Figure 4) show high activity against *S. sclerotiorum*. The EC_50_ was 11.25 and 12.87 μg/mL, respectively, and these values were comparable to the reference compound pyrifenox (EC_50_ = 11.78 μg/mL). The **24c** derivative shows high activity against *B. cinerea* (EC_50_ = 11.97 μg/mL), although it is lower than that obtained for chlorothalonil (EC_50_ = 12.01 μg/mL). In turn, against *R. solani*, compound **24a** is characterized by high activity, higher than pyrifenox (EC_50_ = 9.66 μg/mL vs. 4.02 μg/mL for pyrifenox). 

Aromatic methoxime derivatives have been tested for antifungal activity on phytopathogenic fungi of fruits (Figure 4) [28]. The tests were carried out against five fungal strains: *Penicillum digitatum* CCC-102, *Penicillum italicum* CCC-101, *B. cinerea* CCC-100, *Monilinia fructicola* INTA-SP345 and *Rhizopus stolonifer* LMFIQ-317. It turned out that such compounds do not show satisfactory activity. Of the 14 tested compounds, the most active were the derivatives **25a** and **25b**, which acted on most of the tested strains, although the MIC values were not too high and were in the range of 250–62.5 µg/mL. Only compound **25b** showed promising activity against *R. stolonifer* (EC_50_ = 15.6 μg/mL, comparable to Imazalil). 

Analyzing the effect of the compound’s structure on the antifungal activity, it can be concluded that the dominant group of compounds are derivatives containing nitrogen heterocyclic compounds in their structure, namely, imidazole and triazole. In addition, both in fungicides and compounds with antifungal activity, compounds with a benzyl and phenyl group at the oxygen atom of the oxime ether moiety predominate. Modifications in this group of compounds mainly concern substituents at the carbon atom of the oxime ether moiety.

## 3. The Oxime Ethers with an Antibacterial Activity

The antibacterial drug used in medicine, which contains a carbamoyloxymethyl moiety and belongs to the oxime ethers, is cefuroxime **26** (under the brand name Zinacef) [29], belonging to the group of β-lactam antibiotics and, more precisely, to the second-generation cephalosporins (Figure 5). It is taken orally in the form of a prodrug—cefuroxime acetoxyethyl ester. It is a broad-spectrum antibiotic against both Gram-positive bacteria (e.g., *Streptococcus pyogenes*, *Staphyloccocus aureus*) and Gram-negative bacteria (e.g., *Escherichia coli*, *Haemophilus influenzae*). It is indicated in the treatment of acute bronchitis and infections of the upper respiratory tract and lower urinary tract [29,30]. 

Roxithromycin (**27**, Figure 5) [31,32] is an administered antibacterial macrolide structurally related to erythromycin. It has an in vitro antibacterial profile, with activity against *S. aureus*, *Staphylococcus epidermidis*, *Streptococcus pneumoniae*, *Branhamella catarrhalis*, *Legionella pneumophila*, *Chlamydia trachomatis* and other less-common pathogens. Roxithromycin is characterized by excellent enteral absorption achieving high concentrations in most tissues and body fluids. Clinical efficacy has been confirmed in the treatment of the respiratory tract infections, atypical pneumonias, ear, nose and throat infections and others. Roxithromycin is well tolerated and is an orally active drug which should prove a useful alternative when selecting antibacterial therapy for indications where macrolides are appropriate [31,32]. 

Gemifloxacin **28** belongs to the group of oxime methyl ethers and is a fluoroquinolone derivative with potent in vitro activity against pathogens associated with community acquired the respiratory tract infections. In particular, gemifloxacin has the lowest MIC values against *S. pneumoniae* (MIC_90_ 0.03–0.06 μg/mL) isolates, and the MIC values are not influenced by pneumococal resistance to non-fluoroquinolon antimicrobial agents. In addition, gemifloxacin is not affected by *β*-lactamase production and displays low MIC values against *H. influenzae* (MIC_90_ < 0.008–0.06 μg/mL), *Moraxella catarrhalis* (MIC_90_ < 0.008–0.03 μg/mL), *Haemophilus parainfluenzae*, *S. aureus*, *Chlamydia pneumoniae* and *Mycoplasma pneumoniae* [33]. 

Feng et al. studied benzylic derivatives of gemifloxacin. These derivatives were initially evaluated for their in vitro antibacterial activity against representative Gram-positive and Gram-negative strains. Of the 23 compounds tested, the most active compound **29** (MIC: <0.008–4 mg/mL) was found to be 2–128 times more potent than the parent gemifloxacin against the Gram-positive strains. For example, against *S. aureus*, the ATCC259223 (S.a.) MIC value was <0.008 μg/mL (gemifloxacin 0.06 μg/mL). This compound is less active against the negative bacterial strains [34]. 

The methyl and benzyl ethers of oximes, containing thiophene, piperazine and quinoline groups, also have antibacterial properties. Their activity was tested against Gram-positive strains (*S. aureus*, Methicillin-resistant *S. aureus* (MRSA), *S. epidermidis*, *Bacillus subtilis*) and Gram-negative strains (*E. coli*, *Klebsiella pneumoniae*, *Pseudomonas aeruginosa*). The most active compound **30** showed activity against Gram-positive strains comparable to or higher than the reference compounds: ciprofloxacin and norfloxacin (MIC in the range of 0.19–0.39 µg/mL). The activity of this compound against Gram-negative strains was also the highest among the compounds tested, but the MIC values were higher than those obtained with reference compounds. It was observed that the methyl ethers were characterized by higher antibacterial activity than the benzyl derivatives. It can therefore be concluded that the type of alkoxy group in the oxime ether structure affects the antibacterial activity of the tested compounds [35]. 

Bhandari et al. [36] tested the in vitro antibacterial activity of the benzyl oxime ethers 3,4-dihydronaphthalen-1(2*H*)-one with an imidazolomethyl and triazolomethyl substituent in position 2. The activity was tested against pathogenic strains of bacteria such as *K. pneumoniae*, *E. coli*, *P. aeruginosa*, and *S. aureus*, and it was compared with the standard drug, gentamicin. Studies have shown that compounds containing an imidazole ring were more active than derivatives containing a triazole ring. In the case of structures for which the activity of two geometric isomers was tested, the (*Z*) isomers showed greater antibacterial activity than the (*E*) isomers. Among the 21 tested compounds, the most active (and showing the same activity) were three imidazole derivatives **31**. These derivatives showed better activity against *S. aureus* (MIC = 0.781 µg/mL) and slightly weaker activity against *K. pneumoniae* and *E. coli* (MIC: 1.563 and 0.391 µg/mL, respectively) compared to gentamicin (MIC: 6.25, 0.78 and 0.18 µg/mL, respectively). None of the tested compounds showed activity against *P. aeruginosa* [36]. 

The antibacterial activity of cholesterol derivatives containing the oxime ether group was also studied. Khan et al. [37] obtained six derivatives with an ether group in the 6th position of the steroid system, among which the most active were those containing a chlorine atom in the 3rd position (**32a**,**b**). The compound **32a** showed activity against *S. aureus*, *S. pyogenes*, *Salmonella typhimurium* and *E. coli* comparable to chloramphenicol (MIC = 32 μg/mL). The compound **32b** was equally active against the first two strains, while lower activity was observed against *E. coli* (MIC = 64 μg/mL). 

In turn, Alam et al. [38] studied three derivatives containing an ether group in position 7 of the steroid system. These compounds **33a**–**c** showed activity against the Gram-positive (*B. subtilis*, *S. pyogenes* and *S. aureus*) and the Gram-negative bacteria (*E. coli*, *P. aeruginosa*, *K. pneumonia*) comparable to chloramphenicol. 

Weak antibacterial activity of the benzyl ethers has also been observed in the case of benzofuran-2-yl)ethan-1-one oxime derivatives **34** [39]. 

Kirilmis et al. [40] examined the antibacterial activity of bisbenzofuran-2-yl-methanone oxime ethers against *S. aureus*, *Bacillus megaterium*, *K. pneumonia* and *E. coli*. Among the tested compounds, only ether (**35**) (Figure 5) showed better activity than Streptomycin sulfate against the first three species of bacteria, but it did not show a killing effect on *E. coli*. This compound also showed antifungal activity against *C. albicans*. 

The antibacterial activity is also shown by the oxime ethers containing the (tetrahydroquinoxalin-2-yl)methyl group) [41]. Of the five tested compounds, the most active against *E. coli* and *S. aureus* were propanone and butanone oxime derivatives **36**, for which identical MIC values were obtained, better than for penicillin (62.5 and 125 μg/mL vs. 125 and 250 μg/mL for penicillin, respectively). However, the second reference compound, tetracycline, is much more active against *S. aureus* than the tested compounds (MIC = 3.90 μg/mL). 

Akunuri et al. [42] examined the antibacterial activity of thirty-five (*E*)-1-(1*H*-indol-3-yl) ethanone *O*-benzyl oxime derivatives. Of these, compound **37** showed the best inhibition against *S. aureus* (MIC = 1 μg/mL), albeit four-fold lower than Levofloxacin. In addition, compound **37** has shown potent inhibition against a number of clinically isolated of MRSA and VRSA strains with MIC of 2–4 μg/mL. Other derivatives showed much weaker activity against *S. aureus* and MRSA and VRSA strains. In relation to other tested strains, namely, *E. coli*, *P. aeruginosa*, *Acinetobacter baumannii*, *K. pneumoniae* and *Mycobacterium tuberculosis*, the tested compounds did not show satisfactory activity [42]. 

Ferrocene–oxime ether benzyl 1*H*-1,2,3-triazole hybrids were also tested for activity against the selected Gram-positive and Gram-negative strains. Good activity against *B. subtilis* and *S. aureus* was shown by a derivative containing a fluorine atom in position 4 of the phenyl ring **38**, for which the MIC values were 15.62 μg/mL and 7.81 μg/mL, respectively, and they were comparable to or lower than the reference compound—Streptomycin sulfate. This compound also showed activity against *C. albicans* comparable to fluconazole [43]. 

*O-*Methyloximes of 2,4,6,8-tetraaryl-3,7-diazabicyclo[3.3.1]nonan-9-one derivatives **12** (Figure 2.) have been tested for antibacterial activity against *B. subtilis*, *S. aureus*, *K. pneumoniae* and *P. aeruginosa*. Among the 25 tested oxime ethers, the most active turned out to be those containing one of the following substituents in the 4-position of the phenyl rings: F, CH_3_ or SCH_3_. Among the four bacterial strains, these compounds were the most active against *P. aeruginosa* (MIC_90_ value was 4 µg/mL for all compounds, being four times lower than for Gentamicin) [13]. Among the structural analogs of these compounds, derivatives of 2,4-diaryl-3-azabicyclo [3.3.1]nonan-9-one, the most active against all four strains of the bacteria turned out to be compound **39**, which also contains fluorine atoms in its structure. This derivative showed higher activity than Gentamicin (MIC_90_ was: 4, 2, 2 and 8 μg/mL, respectively). This compound also showed the best fungicidal activity [44]. 

Parthiban et al. [14] also studied the antibacterial activity of the *O*-benzyl oximes ethers, derivatives of piperidin-4-one, tetrahydropyran-4-one and tetrahydrothiopyran-4-one. The compound **40**, a piperidine derivative, turned out to be the most active, which, for all tested strains (*P. aeruginosa*, *S. aureus*, *Salmonella typhi* and *E. coli*), was characterized by MIC values lower than Ciprofloxacin (MIC values of 6.25, 6.25, 25 and 12.5 μg/mL vs. 12.5, 25, 50 and 25 μg/mL for Ciprofloxacin, respectively). The introduction of a methyl group to the nitrogen atom in the piperidine ring did not improve the antibacterial properties of the tested derivatives.

The activity of the benzophenone oxime ethers **41** against bacterial strains (Gram-positive—*Bacillus cereus* (PTCC 1015) and *Staphylococcus aureus* (PTCC 1133); Gram-negative—*P. aeruginosa* (PTCC 1077) and *E. coli* (PTCC 1330)) was tested. The most active against Gram-positive bacteria turned out to be compound **41a**, containing a methyl substituent (MIC = 3.125 μg/mL). However, in relation to Gram-negative bacteria, compound **41b**, containing a benzyl substituent, was the most active (MIC value is 6.25 and 12.5 μg/mL, respectively) [45]. 

Some 1-(2-naphthyl)-2-(imidazole-1-yl)ethanone oxime ethers (**42**, Figure 5) have activity against *S. aureus* and *Enterococcus faecalis*. The most promising were ethyl, benzyl and 4-chlorobenzyl ethers (*S. aureus*) and propyl and 2,4-dichlorophenyl (*E. faecalis*) ethers [16].

The coumarin derivatives containing the oxime ether moiety were tested for activity against *M. tuberculosis* (MTBH37Rv) [46]. Four out of the ten analyzed compounds showed satisfactory anti-tuberculous activity. The MIC values were in the range of 0.04–0.39 µg/mL, with the most active derivative (**43**, Figure 5) having twice the MIC value of isoniazid.

A summary of the antibacterial activity of selected oxime ethers is presented in Figure 6 and in Appendix A. Compound **30** is most active against *S. aureus*. Compounds **30** and **31** showed the greatest activity against Gram-negative bacterial strains *E. coli* and *K. pnaumoniae*. Only compound **40** significantly inhibited the growth of the bacterial resistant strain of *P. aeruginosa*.

Analyzing the structures of the oxime ethers with antibacterial activity, it can be concluded that a significant part of the compounds are derivatives of large-volume ketones (in the structure of ethers there are large substituents at the carbon of the oxime ether moiety). In contrast, the substituents at the oxygen atom of the ether group are smaller, with the methyl or benzyl group dominating here.

## 4. The Oxime Ethers with Antiviral Activity

The oxime ethers have also been tested for antiviral activity. For example, derivatives containing a pyridyl imidazolidinone moiety show very good activity against Enterovirus 71 (EV71). The symptoms for EV71 infections range from nonspecific upper respiratory infection and mild fever to central nervous system infections, particularly viral meningitis, encephalitis and severe myocarditis. EV71(4643) replication is most strongly inhibited by ethyl ether **44** (IC_50_ = 0.001 µM). This compound also shows promising activity against other Enteroviruses and some Coxsackieviruses and Echoviruses (IC_50_ ranging from 0.021 to >25 µM). Promising data were also obtained for methyl ether [47]. 

Barnard et al. [48] tested the activity against picornaviruses (PV) of two pyridazinyl oxime ethers (**45**, Figure 7). Antiviral properties have been tested on rhinoviruses (HRV), belonging to the PV group. These compounds inhibited 56 HRV strains, with IC_50_ values in the range of 0.5–6.70 nM and in many cases comparable to or better than pyrodavil. Studies have shown that these compounds are also active against enteroviruses—Coxsackie (IC_50_ = 773–3.608 nM), ECHO viruses (IC_50_ = 193–5.155 nM) and EV71 (IC_50_ = 1–82 nM). The methyl derivative at position 6 of the pyridazinyl ring also inhibited poliovirus WM-1 (cell line Vero 76) (IC_50_ = 204 nM) and the chlorine derivative of polio virus Chat (cell line Vero 76) (IC_50_ = 82 nM). 

Hepatitis B virus (HBV) inhibitors containing an oxime ether group (**46**, Figure 7) were designed by Tan et al. [49]. The authors tested the ability to inhibit the surface antigens of the hepatitis B virus (HBsAg) and the antigens indicating active replication of the HBV virus (HBeAg) in HepG2.2.15 cells of eighteen derivatives. Compounds **46a** and **46b** were the most promising. IC_50_ values against HBsAg were, respectively, 39.93 and 74.92 µM, and against HBeAg they were 245.96 and 273.87 µM, respectively, and they were better than the results obtained for lamivudine—a commonly used anti-HB drug (IC_50_ for HbsAg = 290.73 µM; IC_50_ for HbeAg = 358.59 µM). The studies also showed that most of the tested compounds showed low toxicity to HepG2.2.15 cells. 

Penta-1,4-diene-3-one oxime ether derivatives containing a quinazolin-4(3*H*)-one scaffold **47** (Figure 7) were evaluated for antiviral activity against tobacco mosaic virus (TMV) [50]. Compounds **47a**, **47b** and **47c** with EC_50_ values of 138.5, 132.9 and 125.6 μg/mL, respectively, showed a high and better effect than in the case of ningnanmycin (EC_50_ = 207.3 μg/mL). The results of the study indicate that these types of derivatives may be of great importance in agriculture, because TMV is a plant virus causing a huge loss of crops, and the agents used so far to combat TMW infection are ineffective and environmentally friendly. 

## 5. Oxime Ethers with Insecticidal, Acaricidal and Antiprotozoal Activity

*O*-Benzyl oxime ethers apart from fungicidal activity, also showed insecticidal activity against *Aphis fabae*. The compound containing the substituent -SCH_3_ (**48**, Figure 7), with LC_50_ = 4.4 mg/L turned out to be the most active. Replacing the sulfur atom with an oxygen atom slightly worsened the activity (LC_50_ = 4.4 mg/L), but both derivatives performed better than chlorfenapyr (19.4 mg/L) and were comparable to the LC_50_ of imidacloprid (4.8 mg/L) [23]. 

Ohsumi et al. [51] examined the insect growth-regulatory activity of 10 oxime ethers containing the *O*-2-(4-phenoxyphenoxy) group **49** (Figure 5) against *Culex pipiens pallens* (common mosquito) and *Musca domestica* (housefly) larvae. The best IC_50_ values against *C. pipiens* showed that the activity of compound **49a** (IC_50_ = 0.00013 × 10^−3^ µmol/L) against *M. domestica* was also very good (IC_50_ = 0.28 µmol/L). Compound **49b** was also characterized by high, albeit almost 10 times lower, activity (IC_50_ = 0.00013 × 10^−3^ µmol/L). Both of these compounds were significantly more active than Methoprene (IC_50_ = 1.5 × 10^−3^ µmol/L). In contrast, the **49c** derivative showed the best activity against *C. pipiens* (IC_50_ = 0.044 × 10^−3^ µmol/L vs. 2.4 × 10^−3^ µmol/L for Methoprene).

An interesting insecticidal effect was also observed for pyrazole oxime derivatives with a 5-trifluoromethylpyridyl group (**50**, Figure 7). These compounds were designed as structural derivatives of fenpyroximate **51**—a compound used in protecting various crops against agricultural mites. Most of these compounds showed good activity against *Plutella xylostella* and *Aphis craccivora* at a concentration of 200 µg/mL. On the other hand, the two most promising derivatives containing 3-fluoro- **50a** and 4-bromophenyl **50b** substituents in the 5th position of the pyrazole ring showed very good activity against *P. xylostella* at a concentration of 50 µg/mL (mortality was 86.42 and 100%, respectively) [52]. 

Oxime ethers were also tested for acaricidal activity. A formulation known and used in the control of important herbivorous mites is Fenpyroximate **51**—a pyrazole-based oxime ether (developed by Nihon Nohyaku Co. Tokyo, Japan, in 1991, [52]). It is used in protecting various crops against agricultural mites. It is effective against Tetranychidae (spider mites), Tarsonemidae, Tenuipalpidae (false spider mites) and Eriophyidae in citrus, apples, pears, peaches, grapes, etc. at a rate of 25–75 g/ha. It is not phytotoxic to fruit, citrus, tea, vegetables and ornamental plants [53]. 

In Liu et al., a series of oxime ether derivatives substituted with 2-methylthio-3′/4′-acetophenone were tested for insecticidal activity against Homopteran and Lepidopteran pests. Compound **52** (Figure 7) had the best activities against *Mythimna separata* and *Nephotettix cincticeps* (LC_50_ was, respectively, 1.6 and 0.78 mg/mL vs. 1.8 and 1.4 mg/mL for Fenvalerate and 10 and 16 mg/mL for Chlorfenapyr) [54].

In a study by Abid et al. [55], oxime ethers, derivatives of 2-acetylpyridine and 2-acetylfuran **53**, showed activity against *Entamoeba histolytica*. In both groups of derivatives, compounds containing morpholine (IC_50_: 0.5 and 0.6 µM, respectively) and piperidine (IC_50_: 1.4 and 1.7 µM), which showed higher activity than metronidazole, were the most active (IC_50_ = 1.9 µM).

Pyrazole oxime derivatives containing a 5-trifluoromethylpyridyl moiety, structural analogs of Fenpyroximate **51** (Figure 7), have been tested for acaricidal activity against *Tetranychus cinnabarinus* [56]. The best activity, comparable to Fenpyroximate (100% mortality at a concentration of 10 µg/mL), was observed for derivatives containing the following substituents in the 5-position of the pyrazole ring: phenyl (**50c**), 3-fluorophenyl (**50a**), 4-bromo- (**50b**) and 4-methoxyphenyl (**50d**).

Dai et al. [56] also studied compounds structurally similar to Fenpyroximate, containing an oxadiazole ring. Of the 21 compounds tested, none showed activity comparable to the reference compounds. For example, compound **54a** showed 100% activity against *T. cinnabarinus* only at 500 μg/mL and was inactive at 100 μg/mL. Compound **54b** showed 100% activity against *Oriental armyworm* at concentrations of 500 and 100 μg/mL, but was inactive at 20 μg/mL. Similar activities were observed with compounds **54a** and **c** against *N. lugens*.

Li et al. [57] studied the activity of 26 oxazoline derivatives containing an oxime ether moiety **55** against *T. cinnabarinus*. These compounds contained various substituents, mainly aromatic and alicyclic. All compounds tested were 100% effective at a concentration of at least 0.01 mg/mL on eggs of *T. cinnabarinus* and at a concentration of 0.001 mg/mL on larvae of *T. cinnabarinus.* In both cases, these values were significantly better than those obtained for etoxazole (100% effectiveness at a concentration of 1 mg/mL).

Isolongifolenone oxime derivatives were tested for their anti-insecticide activity against *soybean aphid* and *rice planthoppers*. Only some compounds showed satisfactory activity. The most active against *soybean aphid* turned out to be the derivative **56a** (reduced rate: 65% at the concentration 200 mg/mL), which showed much lower activity than Flucycloxuron (reduced rate: 100% at the concentration 50 mg/mL). Compound **56b** turned out to be active against *rice planthoppers* (reduced rate: 47% at the concentration 200 mg/mL vs. 100% at the concentration 100 mg/mL for Flucycloxuron) [58]. The compound **56b** also has antibacterial (e.g., *Pneumobacillus*, *Proteus vulgaris* and others) and antifungal (e.g., *Aspergillus niger*, *Colletotrichum musae* and others) activity comparable to reference compounds. 

The methyl ether buparvaquone oxime (**57**) showed activity against *Leishmania donovani* HU3 amastigotes [59]. At a concentration of 30 μg/mL, this compound showed 86.1% inhibition (reference compound—sodium stibogluconate 66.1%), but at a concentration of 10 μg/mL, it showed only 1.9% inhibition (vs. 39.7% for sodium stibogluconate) [59].

Among the antiviral derivatives shown above, an increase in the size of the substituent attached to the carbon atom of the oxime ether moiety results in an increase in the pharmacological effect of the compound.

## 6. The Oxime Ethers with Antidepressive Activity

A well-known drug used in the treatment of depression, registered for the first time in 1983 in Switzerland, is 2-aminoethyl oxime ether—fluvoxamine (**2**, Figure 1). This compound belongs to the group of selective serotonin reuptake inhibitors (SSRIs) [60,61]. In medicine, fluvoxamine is also used in the prevention of unipolar affective disorders, in the treatment of obsessive compulsive disorders, social phobia and anxiety disorders with panic attacks [62]. It has significantly fewer side effects associated with blocking the cholinergic system (e.g., dry mouth) than tricyclic antidepressants (TCAs), and also, it does not adversely affect the myocardium (no cardiotoxic effect) [62]. Recent studies on fluvoxamine also suggest its effectiveness in the treatment of the SARS-CoV-2 virus [63].

Other antidepressants containing an oxime ether group are dibenzocycloheptadiene derivatives: noxiptyline **58** [64,65] and demexiptyline **59** [66] (Figure 8). These compounds belong to the group of tricyclic antidepressants and are characterized by reduced anticholinergic activity and low cardiotoxicity [64].

Over the last 25 years, research has been conducted on the antidepressant activity of many other compounds containing the oxime ether group. For example, ethers of alkyl-aryl ketone oximes and *N*-substituted 2-aminoethanol derivatives have been studied in this direction [67]. Compound **60** turned out to be the most active during the “behavioral despair test”, for which the immobility inhibition value was 96.71%, and it was much higher than that obtained for Fluvoxamine (62.89%).

Welle et al. [68] studied alkyl aryl ketone oxime ethers **61** to increase serotonin levels without inhibiting monoamine oxidase (MAO) and/or norepinephrine (NA) uptake. Their action consisted in a very strong increase in the transmission of serotonin neurons associated with a weaker activation of norepinephrine neurons. The test results showed that the analyzed compounds are devoid of side effects such as MAO inhibition, gastric ulceration or bronchoconstriction and are recommended to patients with depression to improve mood in a daily oral dose of 25–500 mg for adults. The most active were compounds containing a methoxy substituent and a chlorine atom (ED_50_ values for increasing serotonin levels were, respectively: 12 and 15 mg/kg) [68]. Compounds with an analogous structure, differing in the length of the carbon chain in the alkyl part of the starting ketone, increased the concentration of norepinephrine and serotonin but did not inhibit the MAO. The best effect in terms of increasing the level of serotonin was shown by derivatives containing a nitrile **62a** and methyl **62b** substituent (ED_50_: 8.4 and 10 mg/kg, respectively). These compounds are also characterized by very low toxicity and neurotoxicity, and they do not cause gastric ulceration and bronchoconstriction.

De Sous et al. [69] studied *para*-benzoquinone mono-oximes and their oxygenated derivatives for antidepressant effects. An interesting activity was shown by oxime ether **63a** with a structure resembling fluvoxamine, for which the duration of immobility in the mouse was 175.1 s, while for imipramine, which was the reference standard, this value was 178.3 s. Compound **63b**, also containing an oxime ether moiety, also showed a good psychoactive effect (mouse immobilization time was 189.5 s) [69]. 

Nencetti et al. [70] obtained a series of benzyl ethers of oximes derived from 4-arylpiperidin-3-one (**64**, configuration *E*, Figure 8). The final products were tested for inhibition of serotonin (SERT), dopamine (DAT) and norepinephrine (NET) transporters. High serotonin (SERT)-binding affinities (K_i_ = 10.28 nM, fluoxetine 5.80 nM) and high SERT selectivities for the compound **64** have been demonstrated. The compound **64** is inactive towards (NET) and (DAT), and it possesses an affinity for SERT in the same range as fluoxetine and an excellent SERT selectivity.

In the compounds with antidepressant activity, with the exception of derivative **64**, the structural element linking all other derivatives is the unsubstituted and *N*-substituted 2-aminoethyl moiety at the oxygen of the ether oxime group.

## 7. The Oxime Ethers with an Anticonvulsant Activity

Emami et al. [71] evaluated the anticonvulsant activity of (*Z*) and (*E*)-ethers of imidazolylchromanone oxime in a test of convulsions induced by subcutaneous administration of pentylenetetrazol (PTZ) in animals. Seizures caused by the injection of PTZ at 30 mg/kg were significantly delayed by *O*-(2,4-dichlorobenzyl) oximes, but the stereoisomers of (*Z*)-7-chlorochromanone-*O*-(2,4-dichlorobenzyl) oximes **65a** and **65b** (Figure 9) were the most effective, after which seizures occurred only after 715 and 776 s, respectively, and their duration was 40 and 34 s, while for the control group, these values were: 264 (seizure delay) and 52 (seizure duration) seconds. Analyzing the test results, it can be concluded that the substitution of the chlorine atom in the 7-positions and/or the methyl group in the 2-positions of the chroman ring was responsible for the activity of the *O*-(2,4-dichlorobenzyl)oxime derivatives. There were no significant differences in the anticonvulsant effect between the (*E*) and (*Z*) isomers of the tested compounds [71]. 

Alkyl and aryl derivatives of nafimidone oxime ethers [1-(2-naphthyl)-2-(imidazol-1-yl)ethanone, showing antimicrobial activity [16] have also been tested for anticonvulsant activity. Maximum electroshock (MES) and subcutaneous injection of convulsive metrozol (scMet) tests were performed on mice and rats. One of the most promising was oxime ether (**66a**) (Figure 9) with a median effective dose (ED_50_) assessed against MES and scMet-induced seizures of 17.95 and <125 and a median toxic dose (TD_50_) of <150 for intraperitoneal administration to rats [16]. For the derivative with a longer carbon chain (**66b**) in the MES test, the ED_50_ value was 79.51, and in the scMet test, 76.11 mg/kg. 

Compounds with an oxime ether group, which are very strong inhibitors of γ-aminobutyric acid (GABA) and thus have anticonvulsant activity, were described by Knutsen et al. [72]. The inhibition of [3H]-GABA synaptosome uptake was tested in vitro in rats (results are shown in K_i_ values) and tested in vivo in a mouse convulsive model in which seizures were induced by methyl 6,7-dimethoxy-4-ethyl-β-carboline-3-carboxylate (DMCM). Compound **67** turned out to be the most active (K_i_ = 14 nM vs. 67 nM for reference compound) [72]. This compound is an analog of tiagabine, to which the oxime ether function was introduced in place of the CH=CH group.

## 8. The Oxime Ethers with an Inhibition of the β-Receptors

β-Adrenergic receptors are involved in the regulation of the circulatory system. Currently, these include β1-, β2- and β3-adrenergic receptors. Stimulation of β1/β2-adrenergic receptors causes an increase in the frequency and strength of contractions, as well as the speed of diastole. Their stimulation also contributes to increased excitability and faster conduction of action potentials. However, increased stimulation of both types of receptors can lead to arrhythmias. The arrhythmogenic effect of the receptors requires blocking with the use of appropriate antagonists. A recognized antagonism has been demonstrated at β1- and β2-adrenergic receptors. In turn, β3-adrenergic receptors reduce contractility, not in a healthy heart, but in the case of myocardial failure [73].

A drug with an oxime ether group, which is a strong β-adrenergic antagonist, is falintolol (**68**) (Figure 10) [74]. Falintolol is a beta-adrenergic receptor antagonist. Falintolol does not produce any noteworthy side effects and is capable of being an effective beta-blocking agent in open-angle glaucoma therapy [74].

Activity against β1- and β2-adrenergic receptors is demonstrated by derivatives of *O*-(3-alkylamino-2-hydroxypropyl)oxime (**69**) (Figure 10) [75]. A study in isolated guinea pig atria showed that compound **69a** is the most potent β1 receptor antagonist, with 31% inhibition of heart rate and 55% inhibition of atrial contractility. For comparison, the values obtained for propranolol are 70% and 60%, respectively. However, among the 11 tested compounds, only **69b** and **69c** turned out to be β2 receptor antagonists [75].

Angelone et al. [76] described the synthesis of indenopyrazole oxime ethers, which are β1-blockers, and their effect on the work of the heart. The most interesting turned out to be compound **70**, which strongly reduces myocardial contractility and relaxation without interfering with heart rate and coronary pressure, as well as being a good antagonist of β1-adrenergic receptors. Comparing the IC_50_ values relating to the left ventricular pressure for compound **70** and for commonly used β-blockers, it can be seen that a given oxime ether causes an inhibitory effect at a lower concentration (5 × 10^−10^ M) than, for example, propranolol, nadolol or metaprolol (12 × 10^−9^ M) [76]. 

Bai et al. [77] synthesized a new series of dimethoxy-substituted isochroman-4-one oxime ether hybrids whose ability to block β1-adrenergic receptors was tested using an isolated rat left atrium. The best results were obtained for compound **71** (52.2% inhibition in conc. 10^−7^ mol/L and 62.9% in conc. 10^−6^ mol/L). These values for propranolol were 49.7 and 73.8%, respectively. Substituents at the nitrogen atom of the amino group had a significant impact on the β-receptor inhibition power by the tested group of oxime ethers. The replacement of the propyl group in compound **71** with aromatic substituents resulted in a complete loss of the blocking effect. It was also observed that the activity increased with the elongation of the carbon chain of the alkyl group at the nitrogen atom; however, large groups, e.g., *n*-butyl, caused its decrease. Derivatives with an *N*-propyl or *N*-isopropyl group had a high or moderate ability to block β1-receptors, which means that the *N*-alkyl substituent with a three-carbon chain significantly affects the antagonizing properties of the compounds [77].

Tandon et al. [78] synthesized oxime ethers derived from 1-naphthoxepins that could be used as potential hypotensive agents. The effects of the compounds were tested on anesthetized cats and compared with propranolol, which lowered blood pressure by 50 mm Hg over 60 min at a dose of 5 mg/kg and by 46 mm Hg over 40 min at a dose of 1 mg/kg. The best results were obtained with compounds **72** and **73**. Ether **72**, when administered intravenously, decreased blood pressure (BP) by 80 mm Hg over 100 min at the 5 mg/kg dose and by 60 mm Hg over 75 min at the 1 mg/kg. In contrast, derivative **73** at a dose of 5 mg/kg caused a decrease in blood pressure by 80 mm Hg in 16 min, and at a dose of 1 mg/kg by 10 mm Hg in 2 min. Analyzing the data obtained, it can be concluded that oxime ether **72** showed better hipotensive activity than oxime ether **73** and that compound **72** was more potent than the reference drug [78].

Soltani Rad et al. [45] obtained a series of oxime ethers derived from benzophenone and fluorene containing amino acid residue. The most active compound **74** slowed the dog’s heart rate at 20 min from 120 to 90 beats per minute (dose of the substance 2 mg/kg, propranolol as the reference drug caused a change in heart rate from 127 to 105 heartbeats per minute at 20 min of the study (see Figure 10) [45].

Among the presented β-adrenergic receptor inhibitors, a noticeable element connecting all the compounds except for the >C=N-O-R group is the *O*-(3-amino-2-hydroxypropyl) fragment, in which the substituents at the amino group are modified.

## 9. The Oxime Ethers with an Anti-Inflammatory Activity

An oxime ether drug with anti-inflammatory activity is ridogrel (**3**, Figure 1). Its mechanism of action consists of reducing the production of thromboxane B_2_ and reactive oxygen metabolites by the mucosa, as well as inhibiting the activation of platelets [3,79]. 

Anti-inflammatory activity was also shown by 1,2-benzothiazine 1,1-dioxide-3-ethanone oxime *N*-aryl acetamide ether derivatives (**75**) (Figure 11) [80]. Anti-inflammatory activity was tested by measuring the effectiveness of inhibiting the release of tumor necrosis factor (TNF-α), interleukin 8 (IL-8) and monocyte chemotactic protein (MCP-1) in the presence of 4-methoxyamphetamine (PMA)-induced inflammation. The obtained results were compared with the values for piroxicam and pioglitazone. Compound **75a** inhibited both IL-8 (IC_50_ = 16.7 µM) and MCP-1 (IC_50_ = 5.7 µM), while TNF-α was affected by derivatives **75a** (IC_50_ = 12.1 µM) and **75b** (IC_50_ = 6.0 µM). All mentioned oxime ethers had better activity than piroxicam [80]. 

In the work of El-Gamal et al. [81], a number of substituted derivatives of benzylidene acetone oxime ethers (**76**) were synthesized, and then their anti-inflammatory properties were assessed in a study on a rat paw with carrageenan-induced edema. Diclofenac sodium was used as the reference drug. The most promising effects were shown by the oxime ethers **76a** and **76b**, for which the average percentage increase in paw weight was, respectively: 9.70% and 10.71%, these values were almost equivalent to the data for the reference compound (8.12%). The authors observed that compound **76b** also has significant analgesic activity and additionally does not cause ulcer formation [81].

## 10. The Oxime Ethers as the PPAR Agonists

Activation of peroxisome proliferator-activated receptors (PPAR) is important in the treatment of type II diabetes and metabolic syndrome [82]. Good agonists of PPARα and PPARγ receptors turned out to be oxime ethers derived from α-acyl-β-phenylpropanoic acids described by Han et al. [82]. The study was performed on the chimeric PPAR-GAL4 receptor in transfected HepG2 cells. The reference substances were rosiglitazone (EC_50_ for PPARγ = 82 nM) and tesaglitazar (EC_50_ for PPARα = 9798 nM, for PPARγ = 3528 nM). All of the 22 tested compounds showed significantly better activity than the reference drugs, and the most active was *R*-enantiomer of oxime ether **77**, with an EC_50_ for PPARα of 11 nM and for PPARγ of 5 nM, which was also tested for lowering glucose and triglyceride (TG) levels in mouse plasma. Oral administration of racemic mixture of compound **77** at 10 mg/kg resulted in a 73% reduction in glucose and a 58% reduction in triglycerides.

The *O*-methyl oxime ethers the derivatives of benzothiazol-2-one was designed as dual PPARα/γ agonist ligands to the treatment of type 2 diabetes. Compound **78** [83]—(*S*)-enantiomer—tested as a PPARα and PPARγ agonist, obtained for PPARα an EC_50_ of 525 nM (reference drug rosiglitazone 10.00 nM) and for PPARγ an EC_50_ of 0.9 nM (reference drug—rosiglitazone 4 nM). 

Makadia et al. [84] described the PPAR-activating effect of thiazole and oxazole derivatives containing phenoxyacetic acid and an oxime ether group (**79**, Figure 11). The study evaluated the agonistic effect of compounds on PPAR α, γ and δ, and the results were presented as a percentage of the maximum activity of each compound compared to the reference substance (WY-14643 for PPARα, Rosiglitazone for γ, GW-501516 for δ) and normalized to 100 % (E_max_). 

High values of E_max_, equal to 99.8%, 85.3% and 90.1% for PPAR α, γ and δ, respectively, were noted for the thiazole derivative **79a**, and comparing them with the data for GW-501516 (E_max_ = 76.9%, 61.1% and 100.4%, respectively, for PPAR α,γ and δ), it can be concluded that compound **79a** had a much more attractive effect on PPAR α and γ**.** Promising activity (E_max_ = 84.5%, 89.5% and 74.5%) was also shown by the oxime ether **79b**. The study for hypolipidemic and antihyperglycemic properties showed that compound **79** has a better effect—glucose level was reduced by 59.9% and TG by 52.4% [85].

## 11. The Oxime Ethers with an Anticancer Activity

Surkau et al. [85] described oxime ethers based on anthracenone with antiproliferative effects on the cancer cells. The activity was assessed in a study on the K562 human erythroleukemic cell, and the results showed that only some of the compounds effectively inhibited proliferation. The authors suggest that the type of substitution in the terminal phenyl ring as well as in the anthracenone ring influenced the antiproliferative properties. The best, and the one most similar to colchicine (IC_50_ = 0.02 μM), effect was achieved for oxime ether **80a** (IC_50_ = 0.37 μM) and **80b** (IC_50_ = 0.11 μM), which means that the methoxyphenyl group in the 4th position of the phenyl ring is a key group affecting the inhibition of tubulin polymerization in this series of compounds [85].

The antiproliferative activity of a number of pyrazole oxime ether derivatives **81** and **82a-b** was assessed by Park et al. [86]. Compound testing was performed on several human cancer cell lines: HepG2 (liver cancer), MCF-7 (breast cancer), MKN45 (gastric cancer) and A549 (lung cancer). A moderate effect was shown, among others, by compound **81** (Figure 12), which inhibited MCF-7 in 38.2% and A549 in 29.1% at 1 μM [86]. Much more interesting anticancer properties were revealed by derivatives **82** (Figure 12) in the assay against human solid tumor cells of the following lines: A549, SKOV-3 (ovarian cancer), SKMEL-2 (melanoma), XF498 (central nervous system cancer) and HCT15 (colorectal cancer). The reference drugs were cisplatin (IC_50_, respectively: 3.09, 3.42, 3.28, 3.47 and 6.91 μg/mL) and doxorubicin (IC_50_: 0.05, 0.09, 0.07, 0.09 and 0.28 μg/mL, respectively). Compounds **82a** and **82b** had the most attractive cytotoxic abilities—better than reference substances, they affected XF498 (IC_50_: 0.04 and 0.02 μg/mL, respectively) and HCT15 (IC_50_: 0.02 and 0.01 μg/mL), while A549 (IC_50_: 0.12 and 0.10 μg/mL) and SKOV-3 (IC_50_: 0.21 and 0.28 μg/mL) will perform better than cisplatin [86]. 

The anticancer properties are also possessed by oxime ethers of ecdysteroid derivatives, which were tested in the work of Vágvölgyi et al. [87]. For example, compound **83** inhibits the proliferation of HeLa cells (cervical cancer, IC_50_ = 8.43 μM) and MDA-MB-231 (breast cancer, IC_50_ = 12.36 μM) more than cisplatin (IC_50_: 14.02 μM and 18.65 μM, respectively) [87]. 

The anticancer activity of androstane estrane oxime ethers was tested in vitro on a panel of 60 cell lines. Among the 21 analyzed compounds, the most satisfactory results were obtained for derivatives **84**, **85** and **86** (Figure 12). The log GI_50_ (GI_50_—drug concentration that resulted in a 50% reduction in protein growth) was −4.99, −5.35 and −5.66, respectively, the log TGI (TGI—drug concentration that completely inhibited growth) was −4.55, −4.77 and −5.21, and the log10LC_50_ (LC_50_—drug concentration causing a 50% decrease in the measured protein): −4.13, −4.23 and −4.76 [88]. 

Steroid derivatives, due to their potential anti-cancer properties, have been used to develop new compounds—androst-4-ene oxime ethers with a built-in pyridine ring [89]. The compounds were tested against eight human cancer cell lines. The oxime ether **87** showed very high activities against A-549, HT-29 and MDA-MB-231 (breast cancer cell line), and IC_50_ values were 2.0, 3.3 and 4.7 μM, respectively, after 72 h incubation (activity values for reference drug—cisplatin—were 3.2, 4.1 and 2.6 μM, respectively) [89]. 

The aim of the study by Berényi et al. [90] was to test the antiproliferative properties of the synthesized estrone-16-oxime ethers (**88**) (Figure 12) on human cell lines isolated from the following cancers: cervical (HeLa), ovarian (A2780), breast (MCF-7) and epidermis (A431); and on non-cancerous lung fibroblasts (MRC-5). Compound **88** showed a very good effect on HeLa (IC_50_ = 5.63 µM), A431 (IC_50_ = 13.25 µM) and MRC-5 (IC_50_ = 6.94 µM). The activity values for the reference drug, cisplatin, were 5.66, 8.81 and 4.13 µM, respectively [90]. 

In the publication of Díaz et al. [91], the cytotoxicity of the oxime ethers derived from flavone *O*-alkyloximes and 6-hydroxyflavone (**89**) was synthesized and tested (Figure 12). Anti-proliferative activity was assessed against MDA-MB-231 (breast cancer), PC-3 (prostate cancer), A-549 (lung adenocarcinoma) and MRC-5 (lung derived fibroblasts) cells. Among the tested oxime ethers, those containing the hydroxyl substituent in position 6 of the flavonoid system turned out to be active. They acted mainly on the MDA-MB-231 line, and one of the better IC_50_ values relating to the inhibition of the proliferation of these cells, albeit much higher than the reference drug—Vincristine (IC_50_ = 0.008 μM)—was noted for **89a** (IC_50_ = 28.7 μM) and **89b** (IC_50_ = 40.4 μM) [91]. 

Synthesized derivatives of naringenin oxime (**90**) ethers (Figure 12) have also been tested for antiproliferative properties [92]. The activity was assessed on human cell lines: acute promyelocytic leukemia (HL-60), cells isolated from cervical cancer (HeLa, Siha) and breast cancer (MCF-7, MDA-MB-231). The most sensitive to the tested compounds were HeLa and MCF-7 cells, and the most proliferation inhibiting was the derivative of *tert*-butyl oxime ether (**90a**), which, at a concentration of 50 µM, stopped the growth of all tested cell lines at a rate of 87–92.22% (this was similar to cisplatin, where the range of values at the same concentration was 84.43–99.09%). MCF-7 was moderately affected by the benzyl derivative (**90b**) (growth inhibition at 50 µM was 64.47%) [92]. 

In the publication by Chakravarti et al. [93], the anticancer properties of thioarylnaphthyl methanone oxime ether analogs were investigated. The antiproliferative effect has been tested on the human cancer cell lines such as: MCF-7 (hormone receptor positive breast cancer), MDA-MB-231 (ER negative breast cancer), DU-145 (ER positive prostate cancer), Ishikawa (ER-positive endometrial adenocarcinoma) and HeLa (cervical cancer). The best EC_50_ values against all tested cells were obtained for compounds **91a** (EC_50_ values in the range of 5.01–9.5 µM) and **91b** (EC_50_ values in the range of 7.2–12 µM. Moreover, oxime ether **91b** at a dose of 16 mg/kg reduced tumor growth in MCF-7 xenograft mice [93].

A series of 21 new penta-1,4-dien-3-one oxime ethers containing a quinazoline ring were obtained by Su et al. [94]. The compounds were tested for their antitumor activity. The results showed that most of the compounds had inhibitory effects (exhibited extremely inhibitory effects against hepatoma SMMC-7721 cells). In particular, compounds **92a** and **92b** (Figure 12) showed high activity with IC_50_ = 0.64 μM and 0.63 μM (against the reference drug gemcitabine—1.40 μM) [94]. 

Bisphenol **93** derivative was identified as a novel estrogen receptor (ER) agonist and exhibited moderate anti-proliferative activities on MCF-7 cells, but it did not induce cancer cell proliferation. EC_50_ for this compound (**93**) was 0.075 μM, indicating that they have significant estrogen responsive element (ERE) mediated transcriptional activity [95]. Kosmalski et al. [96] screened the oxime ethers with heterocyclic, alicyclic and aromatic moieties for their cytotoxicity. The most potent and specific compound was (*E*)-1-(benzothiophene-2-yl)ethanone *O*-4-methoxybenzyl oxime (**94**), which was selective for HeLa (with EC_50_ = 27.9 μg/mL) cells.

Analyzing the results of the antiproliferative activity of the selected oxime ethers presented in Figure 13 and Appendix A, the most promising compound with antiproliferative activity are compounds **91a**,**b**, among which **91b** proved to be effective in in vivo studies. Additionally, ether oxime **93** is also notable for its high activity against the MCF-7 breast cancer cell line.

## 12. Oxime Ethers with Other Biological Activities

### 12.1. The Oxime Ethers with an Antioxidant Activity

The steroid oxime ether 7-(2′-aminoethoxyimino)cholest-5-ene (**95**) and its derivatives with antibacterial activity, synthesized by Alam et al. [38], were also tested for antioxidant activity with using the DPPH test (evaluates the ability to remove the diphenyl-2-picrylhydrazyl radical). All the tested compounds showed worse effects than the ascorbic acid used as the standard (IC_50_ = 5.40 µg/mL), but the most comparable IC_50_ values had the **95** derivative (Figure 14) (IC_50_ = 5.61 µg/mL). 

The antioxidant activity was also demonstrated by derivatives of naringenin oxime ethers, which also had antiproliferative properties and were tested by Latif et al. [92]. Antioxidant activity was assessed by DPPH, ORAC (oxygen radical absorption) and xanthine oxidase inhibition tests. The most satisfactory data were collected for the methyl substituted ether **96**. It showed the highest antioxidant capacity in the DPPH test (EC_50_ = 212.20 µM) and was the only one to be stronger than rutin (standard, 12.35 µmolITE/µmol) in the ORAC test (16.63 µmolITE/µmol) [92].

### 12.2. The Oxime Ethers with an Antiulcer Activity 

The gastric hydrogen-potassium ATPase (H+-K+-ATPase) is a proton pump responsible for the final phase of releasing hydrochloric acid into the lumen of the stomach. Blocking the hydrogen-potassium ATPase by proton pump inhibitors leads to irreversible inhibition of this enzyme [97]. 

Wu et al. [98] obtained bisabolangelone oxime ether derivatives (**97**) and tested them for H+/K+-ATPase inhibition. All tested compounds showed higher activity than omeprazole (IC_50_ in the range of 17~46 µM vs. 80 µM for omeprazole). The highest proton pump inhibitory activity was shown by derivatives **97a** and **97b** (Figure 14), whose IC_50_ were 17.31 and 22.66 µM, respectively.

### 12.3. The Oxime Ethers with an Antiaggregation Activity

Varache-Lembege et al. [99] synthesized methoxyphenyl thienyl ketoxime ethers and studied their inhibitory effect on platelet aggregation induced by arachidonic acid (AA). Compounds (Z)-**98a** and (Z)-**98b** (IC_50_ = 55 mM) was four times more active than acetylsalicylic acid (ASA) (IC_50_ = 225 mM). The (*Z*)-isomer of oxime ether **98c** (IC_50_ = 110 µM) was twice as strong as ASA. Derivatives containing a methoxy group in the *ortho* and *meta* position were much less active, while the (*E*) isomers were characterized by much weaker anti-aggregation properties compared to the (*Z*) isomers [99]. 

### 12.4. Compounds with Monoamine Oxidase (MAO) and Acetylcholinesterase (AChE) Inhibitory Activity 

Cholinesterase inhibitors are used in the treatment of Alzheimer’s disease (AD) in palliative care settings, and numerous studies have shown that monoamine oxidase inhibitors (MAO) can halt or delay the progression of AD. Blocking MAO-A is used in the treatment of depression and anxiety, and MAO-B in the treatment of Parkinson’s disease and AD. Min Oh et al. [100], considering that oxime ethers have a wide range of biological activities, and ethyl acetohydroxamate chalcones inhibit MAO-B and acetylcholinesterase (AChE), studied 24 new chalcone oxime ethers (**99**). The vast majority of these compounds showed selective MAO-B inhibition. The ethers **99a** and **99b** had the strongest effect on MAO-B, and the IC_50_ values were, respectively: 0.018 and 0.028 µM. Compound **99a** showed comparable activity to pargyline (IC_50_ = 0.020 mM). The oxime ether **99c** inhibited both AChE (IC_50_ = 4.39 µM) and MAO-B (IC_50_ = 0.028 µM) well [100].

### 12.5. The Oxime Ethers with an Herbicidal Activity

The herbicidal properties were tested for a series of 12 benzophenone oxime ether derivatives with a secondary amine group (**100**) (Figure 14) [101]. The activity of the compounds was tested on *Oryza sativa*, *Sorghum sudanense*, *Brassica chinensis and Amaranthus mangostanus* L., observing the growth of their root, and tribenuron was used as a standard, with IC_90_ against the tested plants, respectively: 12.28, 13.70, 11.89 and 11.06 mg/L. All tested compounds turned out to be good herbicides. The best effect to *B. chinensis* and *A. mangostanus* L. was shown by the derivative **100a** (IC_90_ = 16.83 and 11.72 mg/mL, respectively). Derivative **100b** showed the highest activity to *S. sudanense* (IC_90_ = 11.27 mg/L), compound **100c** to *Oryza sativa* (IC_90_ = 13.70 mg/mL), and compound **100d** to *A. mangostanus* L. (IC_90_ = 11.22 mg/mL) [101]. 

## 13. Conclusions

The oxime ethers are oxime derivatives with a built-in ether function. These compounds can be described by the general formula: R(R^1^) > C=N-O-R^2^. This is an interesting group of compounds due to the fact that the formation of an oxime ether moiety is one way to combine different structural elements into one molecule. This gives the possibility of composing compounds that exhibit various biological activity. The oxime ether moiety is present in many drugs, e.g., siponimod, ridogrel, fluvoxamine and oxiconazole. Oxime ethers show significant activity, e.g., against Gram-positive and Gram-negative bacteria, which may lead to the discovery of another (in addition to the currently used cefuroxime, roxithromycin and gemifloxacin) antibiotic containing the oxime ether group. The literature also describes many oxime ether compounds with diverse biological activity. The presence of the oxime ether group in drugs used in various diseases is an incentive to search for new compounds containing this structural fragment and to study their various biological activities. These compounds have been tested for their antimicrobial, anticonvulsant, anti-inflammatory, antidepressant, anticancer and other activities. 

This publication presents an overview of the oxime ethers and describes their various biological activities. To the best of our knowledge, this is the first review of the biological activity of compounds containing such a moiety. The authors hope that this review will inspire scientists to take a greater interest in this group of compounds, as it constitutes an interesting research area.

## Figures and Tables

**Figure 1 molecules-28-05041-f001:**
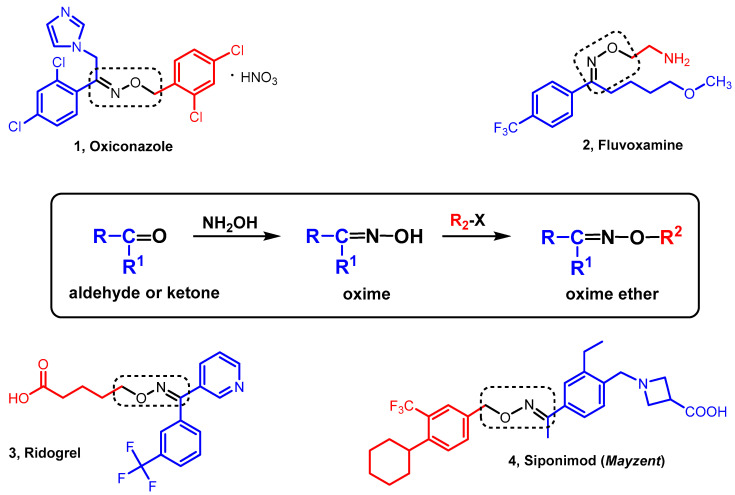
Examples of drugs with an oxime ether moiety.

**Figure 2 molecules-28-05041-f002:**
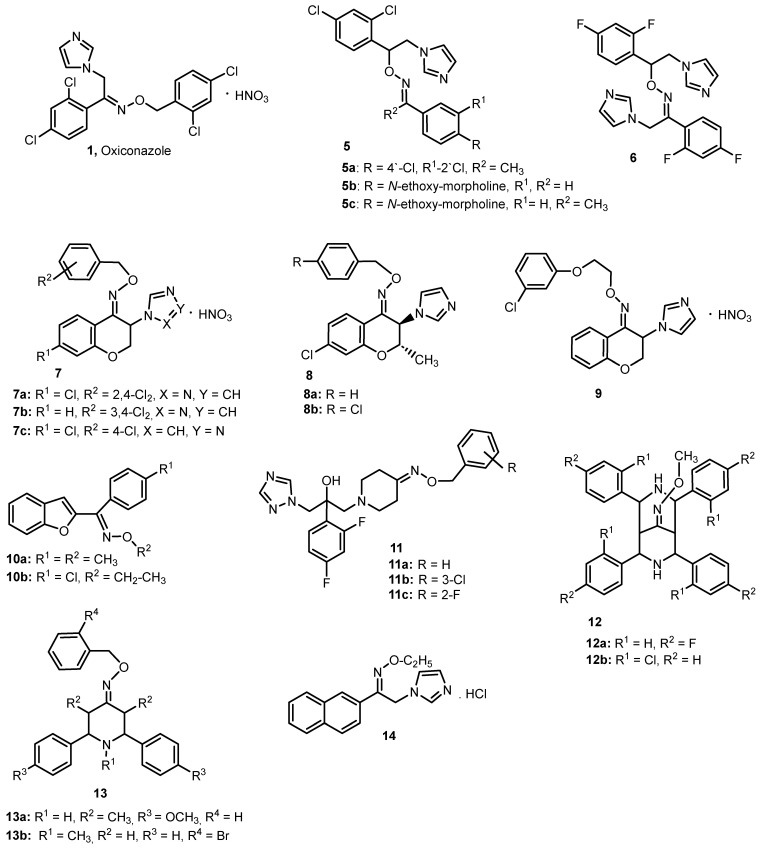
The structures of the oxime ethers **1**,**5**–**14** with the antifungal activity.

**Figure 3 molecules-28-05041-f003:**
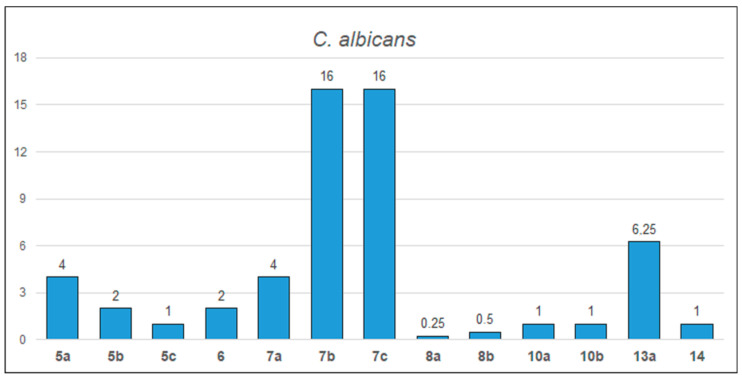
Minimal inhibitory concentrations (MIC, µg/mL) of the selected oxime ethers against *C. albicans* fungal strain.

**Figure 4 molecules-28-05041-f004:**
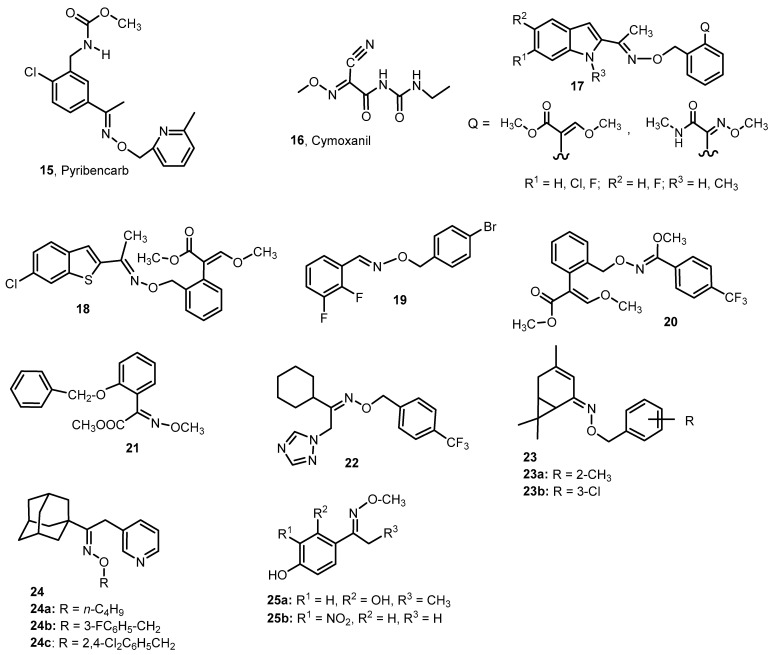
The structures of the oxime ethers **15**–**25** with the fungicidal activity.

**Figure 5 molecules-28-05041-f005:**
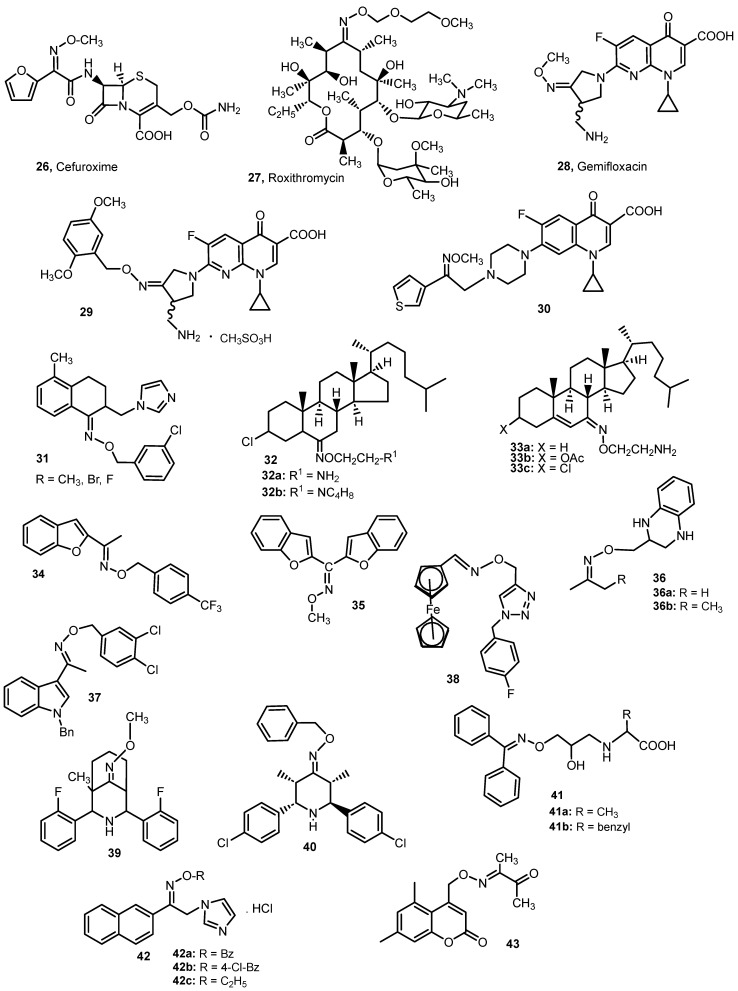
The structures of the oxime ethers **26**–**43** with the antibacterial activity.

**Figure 6 molecules-28-05041-f006:**
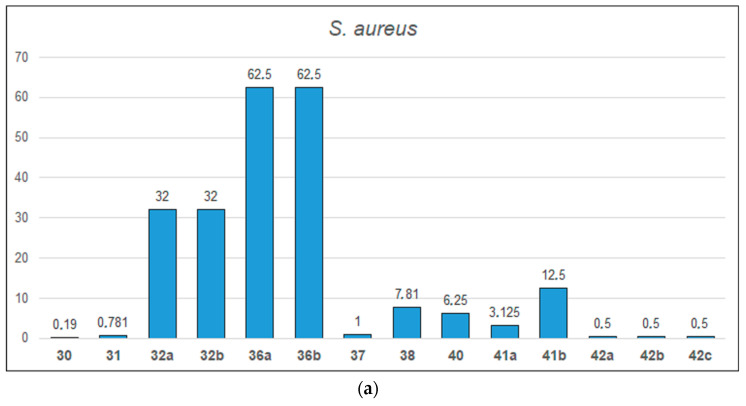
Minimal inhibitory concentrations (MIC, µg/mL) of the selected oxime ethers against (**a**) *S. aureus* and (**b**) *E. coli* bacterial strains.

**Figure 7 molecules-28-05041-f007:**
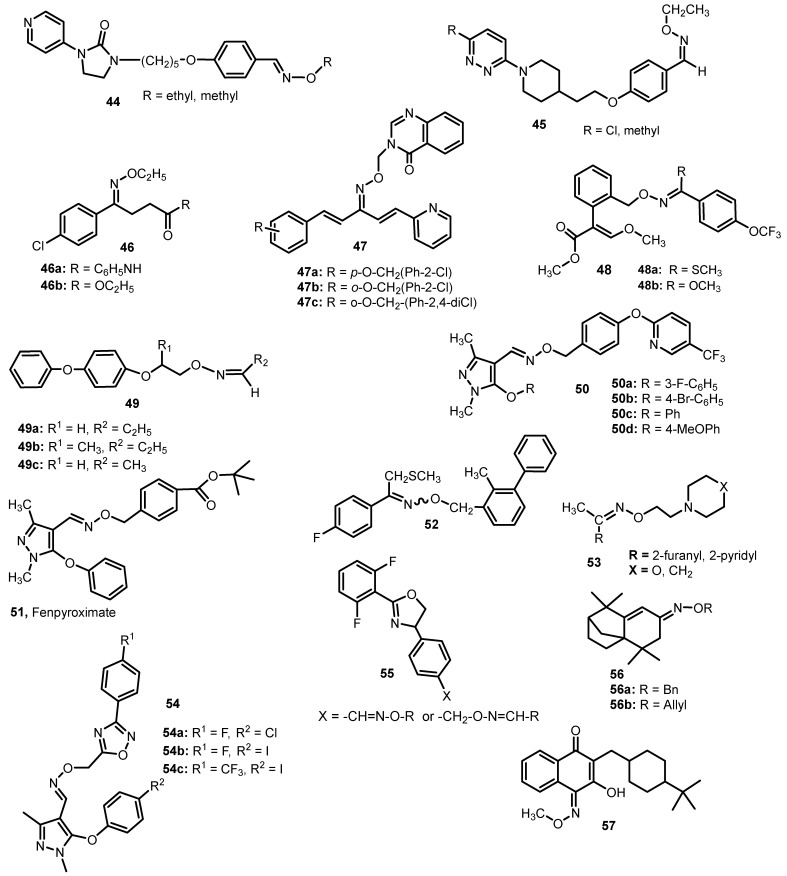
The structures of the oxime ethers with antiviral (**44**–**47**), insecticidal and acaricidal activity (**48**–**57**).

**Figure 8 molecules-28-05041-f008:**
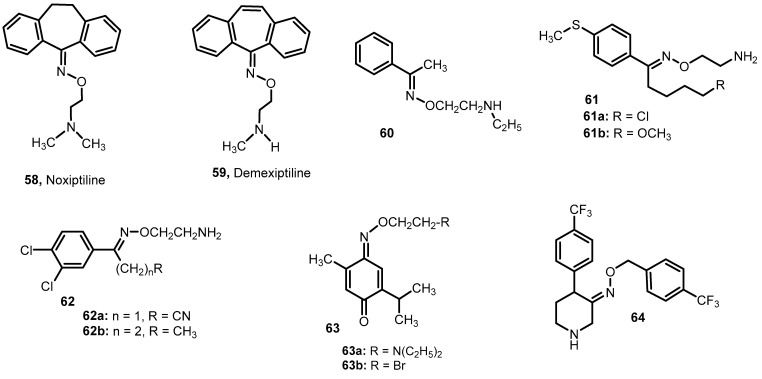
The structures of oxime ethers **58**–**64** with the antidepressive activity.

**Figure 9 molecules-28-05041-f009:**
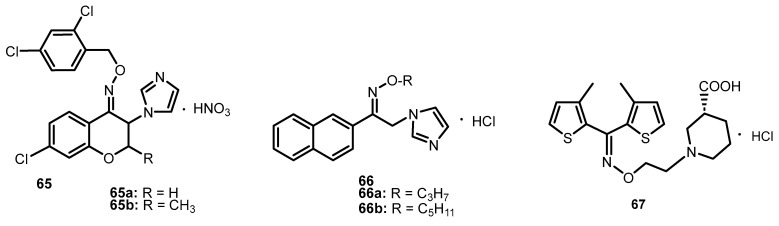
The structures of the oxime ethers **65**–**67** with the anticonvulsant activity.

**Figure 10 molecules-28-05041-f010:**
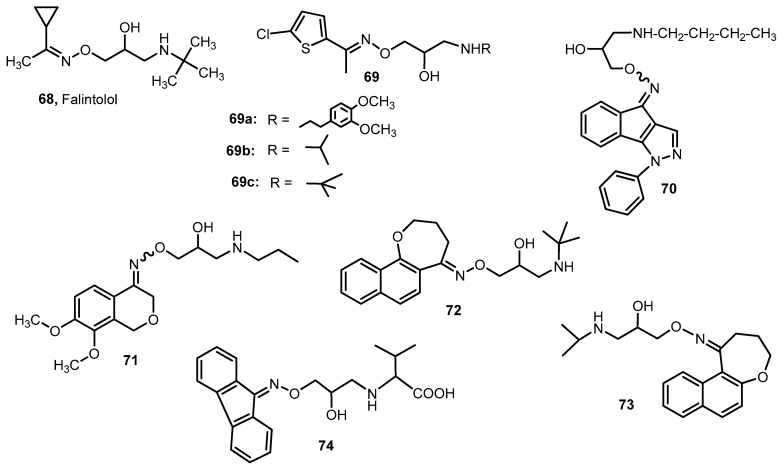
The structures of the oxime ethers **68**–**74** with the antiulcer and antiadrenergic activity.

**Figure 11 molecules-28-05041-f011:**
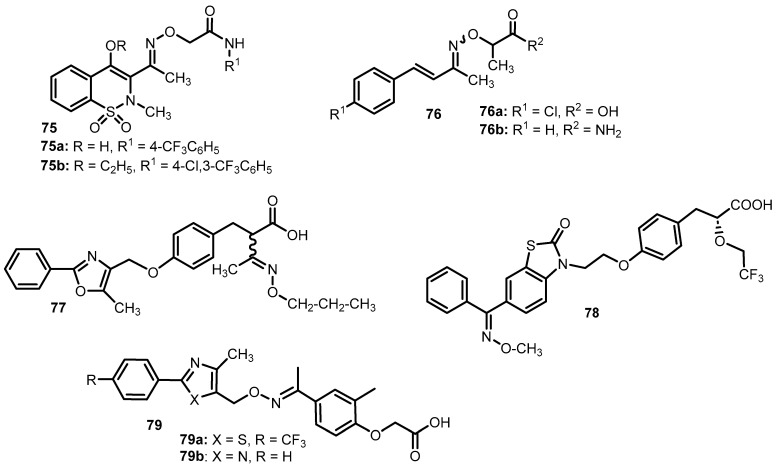
The structures of the oxime ethers with the anti-inflammatory activity (**75**–**76**) and the PPAR agonists (**77**–**79**).

**Figure 12 molecules-28-05041-f012:**
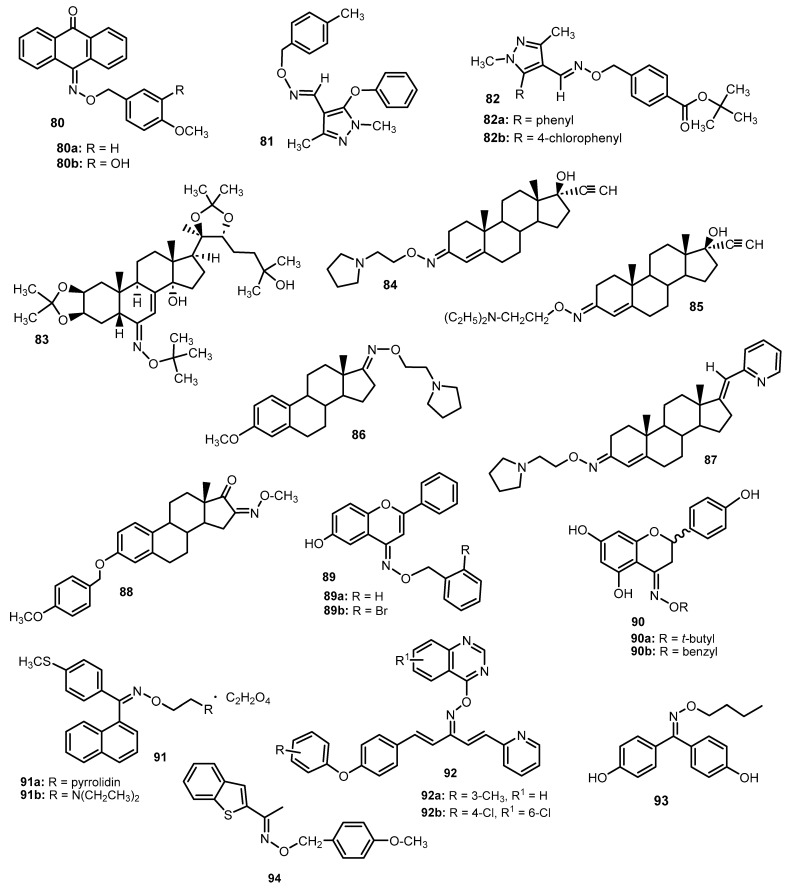
The structures of oxime ethers **80**–**94** with the anticancer activity.

**Figure 13 molecules-28-05041-f013:**
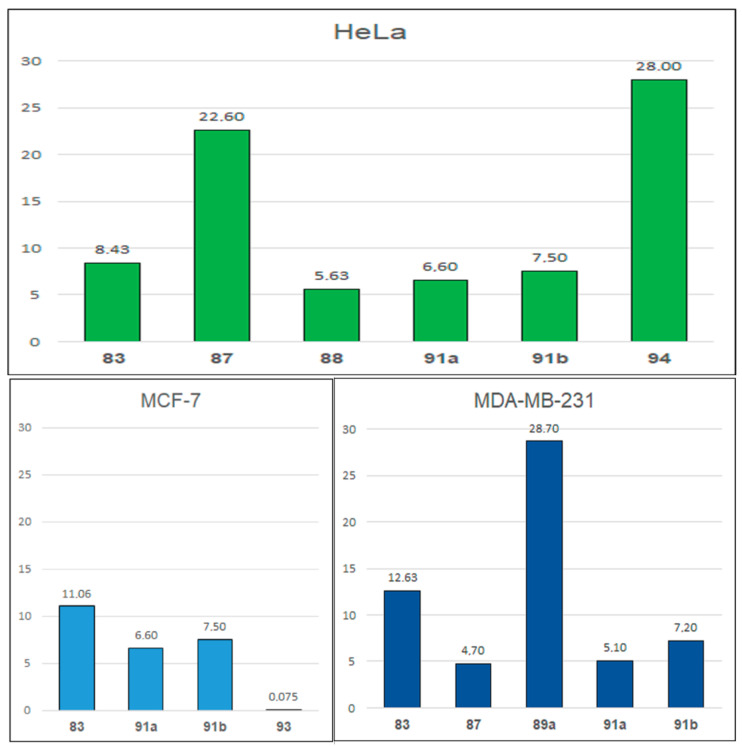
Cytotoxicity (IC_50_, µM) of the selected oxime ethers against selected human cancer cell lines: HeLa, MCF-7 and MDA-MB-231.

**Figure 14 molecules-28-05041-f014:**
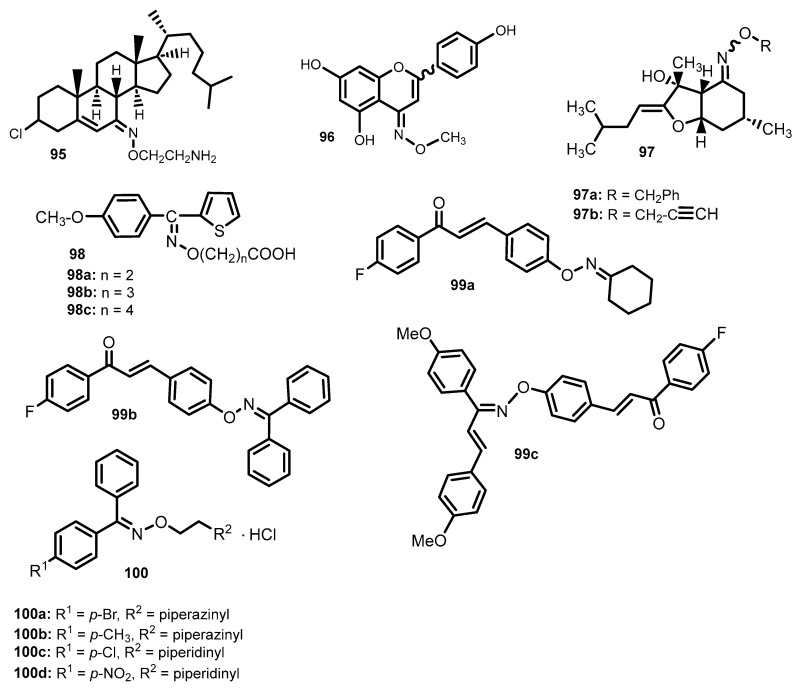
The structures of the oxime ethers **95**–**100** with the antioxidant, antiulcer, anti-AChE and herbicidal activities.

## Data Availability

Data sharing not applicable.

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
