# Peer review of "A Review of Biologically Active Oxime Ethers"

_molecules, 2023, doi:10.3390/molecules28135041_

Round 1

Reviewer 1 Report

Manuscript entitled “A review of biologically active oxime ethers” could be interesting for the readers. This paper needs a minor revision before publication. I have listed a few comments that need to be addressed:

1.       Improve the introduction part with more background about the work.

2.       What is the novelty of this review work that should be clearly discussed at the end of introduction? Also, cite recent review articles on this topic in the introduction.

3.       Write the full form once when mentioning for the first instance.

4.       The structure of this is not appropriate. There should be more information regarding the oxime ethers such as chemistry, structure, etc. then biological activity. Revise it carefully.

5.       The integration of the results from different parameters should be improved carefully.

6.       The application part could be better, improve it carefully with more information. I would recommend to add one or two Tables and Figures in this section based on the specific application.

7.       Conclusion could be better. Also, add limitations and future perspectives.

8.       Add up-to-date reference and remove old references.

9.       Also, carefully revise the typos and linguistic errors to make the manuscript error-free.

Author Response

We would like to thank the Reviewer for the valuable remarks and comments, which significantly helped us to improve our manuscript. Below we present point-by-point responses to the comments.

Manuscript entitled “A review of biologically active oxime ethers” could be interesting for the readers. This paper needs a minor revision before publication. I have listed a few comments that need to be addressed:

  1. Improve the introduction part with more background about the work.

It has been corrected in manuscript.

  1. What is the novelty of this review work that should be clearly discussed at the end of introduction? Also, cite recent review articles on this topic in the introduction.

We have made a thorough review of the available literature and have not found any older reviews describing the biological activity of oxime ethers. To the best of our knowledge, this is the first review of the activity of compounds containing such a moiety. There are earlier publications on the synthesis of this group of compounds, as well as their use in organic synthesis (eg. Vessally et al. Journal of the Iranian Chemical Society 13, pp. 1235–1256 (2016), Sukhorukov et al. Chem. Rev. 2011, 111, 8, 5004–5041, Mirjafary et al. RSC Adv., 2015,5, 79361-79384, Mirjafary et al. RSC Adv., 2016,6, 17740-17758). However, they are outside the scope of this review, and therefore are not cited.

  1. Write the full form once when mentioning for the first instance.

The changes have been applied. Names, that appear for the first time are full. Again, they are in abbreviated form.

  1. The structure of this is not appropriate. There should be more information regarding the oxime ethers such as chemistry, structure, etc. then biological activity. Revise it carefully.

It has been corrected in manuscript

  1. The integration of the results from different parameters should be improved carefully.

The data collected in this review is very diverse in terms of the research techniques used, the parameters tested, the doses and concentrations of the analyzed compounds, which is why it is difficult to unify it more. However, the authors tried to improve the readability of the manuscript by organizing the divergent results regarding, eg. anticancer, antibacterial or antifungal activity in the form of graphs and tables in Supplementary Materials.

  1. The application part could be better, improve it carefully with more information. I would recommend to add one or two Tables and Figures in this section based on the specific application.

It has been corrected in the manuscript.

  1. Conclusion could be better. Also, add limitations and future perspectives.

It has been corrected in the manuscript.

  1. Add up-to-date reference and remove old references.

This has been corrected in the Introduction section

  1. Also, carefully revise the typos and linguistic errors to make the manuscript error-free.

The typos and linguistic errors has been corrected.

Reviewer 2 Report

The manuscript entitled “A review of biologically active oxime ethers” shows a comprehensive review about some oxime ether derivatives and their biological activities. It displayed plenty of compounds containing oxime moiety along with their biological activities. There are some points that should be considered to improve the quality of this article.

-        Add a short paragraph under each category explaining the collective structural activity relationship (SAR) of these derivatives.

-        There are some linguistic and writing errors should be corrected and the whole manuscript should be reviewed for this kind of mistakes like:

-        Line 34: An derivative changed to a derivative

-        Line 40: benzyl changed to a benzyl

-        Line 50: activity changed to activities

-        Line 112 suggest changed to suggested

-        Line 879 a space between the full stop and all

-        Some errors in the figures include:

-        Figure 2: The 1-4 changed to (1, 5-14)

-        Figure 3: Structure no 17: correct this structure

-        Figure 5: Compound 48 can be divided into 48a, 48b like others

-        Figure 7: Adjust the HCl position in compound 67

-        Figure 9: Put compounds 76a, 76b in one compound to unify the method of drawing and remove 3 from the caption.

-        Figure 10: Remove the H atom from the phenyl ring of compound 81 and adjust the bond angles of compound 94

-        Figure 11: Adjust the position of HCl in compound 100 and put 99a, 99c in one compound to unify your way of drawing.

There are some linguistic and writing mistakes need to be corrected.

Author Response

We would like to thank the Reviewer for the valuable remarks and comments, which significantly helped us to improve our manuscript. Below we present point-by-point responses to the comments.

The manuscript entitled “A review of biologically active oxime ethers” shows a comprehensive review about some oxime ether derivatives and their biological activities. It displayed plenty of compounds containing oxime moiety along with their biological activities. There are some points that should be considered to improve the quality of this article.

-        Add a short paragraph under each category explaining the collective structural activity relationship (SAR) of these derivatives.

It has been added to the manuscript.

-        There are some linguistic and writing errors should be corrected and the whole manuscript should be reviewed for this kind of mistakes like:

-        Line 34: An derivative changed to a derivative

-        Line 40: benzyl changed to a benzyl

-        Line 50: activity changed to activities

-        Line 112 suggest changed to suggested

-        Line 879 a space between the full stop and all

The linguistic and writing errors has been corrected.

-        Some errors in the figures include:

-        Figure 2: The 1-4 changed to (1, 5-14)

-        Figure 3: Structure no 17: correct this structure

-        Figure 5: Compound 48 can be divided into 48a, 48b like others

-        Figure 7: Adjust the HCl position in compound 67

-        Figure 9: Put compounds 76a, 76b in one compound to unify the method of drawing and remove 3 from the caption.

-        Figure 10: Remove the H atom from the phenyl ring of compound 81 and adjust the bond angles of compound 94

-        Figure 11: Adjust the position of HCl in compound 100 and put 99a, 99c in one compound to unify your way of drawing.

Most of the figures have been corrected. We only left unchanged the compounds 99a-c to maintain the readability of the structures.

Comments on the Quality of English Language

There are some linguistic and writing mistakes need to be corrected.

The typos and linguistic errors has been corrected.

Reviewer 3 Report

The review, which is entitled "A review of biologically active oxime ethers" pretty much delivers what the title says. The authors view a wide range of oxime ethers and discuss their biologicall/medicinal properties. The review is sound and useful, but I do question whether the oxime ether component of these structures is the part that delivers these properties.The structures are diverse indeed, including ferrocenes, imidazoles, pyridines, benzothiophenes, etc. The writing is mostly fine and the review is well organized. I have only one real criticism. Many of the structures are chiral but in most cases the stereochemistry is not shown. This applies to the whole review but I will just discuss Figure 4. Structure 26 has a chiral center but the stereochemistry is not shown. The same applies to structures 28, 29 31, and 36. Structures 30 and 31 are steroids and the stereochemistry will be dictated by the source. Nevertheless, the stereochemistry should be indicated. Structure 40 has 4 stereogenic centers and therefore could give 16 stereoisomers. Structure 41 has 2 chiral centers and therefore may give 4 stereoisomers. The stereochemical identity of bicyclic derivative is also not shown. And this is just Figure 4. I strongly recommend that the authors address the stereochemistry of all of the structures shown in this review. After all, the chiral nature of these structures will have a profound impact on their biological properties.

Slightly idiosyncratic in places but otherwise fine.

Author Response

We would like to thank the Reviewer for the valuable remarks and comments, which significantly helped us to improve our manuscript. Below we present point-by-point responses to the comments.

The review, which is entitled "A review of biologically active oxime ethers" pretty much delivers what the title says. The authors view a wide range of oxime ethers and discuss their biologicall/medicinal properties. The review is sound and useful, but I do question whether the oxime ether component of these structures is the part that delivers these properties.The structures are diverse indeed, including ferrocenes, imidazoles, pyridines, benzothiophenes, etc.

We thank the reviewer for this important point. We have made the appropriate correction in the Introduction section.

The writing is mostly fine and the review is well organized. I have only one real criticism. Many of the structures are chiral but in most cases the stereochemistry is not shown. This applies to the whole review but I will just discuss Figure 4. Structure 26 has a chiral center but the stereochemistry is not shown. The same applies to structures 28, 29 31, and 36. Structures 30 and 31 are steroids and the stereochemistry will be dictated by the source. Nevertheless, the stereochemistry should be indicated. Structure 40 has 4 stereogenic centers and therefore could give 16 stereoisomers. Structure 41 has 2 chiral centers and therefore may give 4 stereoisomers. The stereochemical identity of bicyclic derivative is also not shown. And this is just Figure 4. I strongly recommend that the authors address the stereochemistry of all of the structures shown in this review. After all, the chiral nature of these structures will have a profound impact on their biological properties.

We thank the Reviewer for this valuable comment. We fully agree with the reviewer that the chiral nature of the compounds is important and has a profound impact on their biological properties. Therefore, wherever the stereochemical aspect was indicated in the source references, it was included in this review. In other cases, the cited references did not indicate which stereoisomers were tested, and whether they were isolated or a mixture of stereoisomers was tested. There we left the structures of compounds in the form in which they are given in the source works. We believe that making changes to the structure of the compounds would be too much interference in the sources references and could cause unnecessary confusion for the reader.

Comments on the Quality of English Language

Slightly idiosyncratic in places but otherwise fine.

The linguistic and writing errors has been corrected.

Reviewer 4 Report

The manuscript entitled “A review of biologically active oxime ethers” by Tomasz Kosmalski and etal collected and presented the structures of oxime ethers with specific bio logical activity. The review includes both those substances that are currently used as drugs (e.g. fluvoxamine, mayzent, ridogrel, oxiconazole), as well as non-drug structures for which various biological activity studies have been conducted.   

In my perspective, the manuscript holds value for readers and is appropriate for publication once it undergoes revision to rectify grammatical and typographical errors.

Author Response

We would like to thank the Reviewer for the valuable remarks and comments, which significantly helped us to improve our manuscript. Below we present point-by-point responses to the comments.

The manuscript entitled “A review of biologically active oxime ethers” by Tomasz Kosmalski and etal collected and presented the structures of oxime ethers with specific bio logical activity. The review includes both those substances that are currently used as drugs (e.g. fluvoxamine, mayzent, ridogrel, oxiconazole), as well as non-drug structures for which various biological activity studies have been conducted.  

In my perspective, the manuscript holds value for readers and is appropriate for publication once it undergoes revision to rectify grammatical and typographical errors.

We thank the reviewer for his favorable review of our manuscript. All grammatical and typographical errors have been corrected.

Round 2

Reviewer 1 Report

The manuscript can be accepted for publication.

Reviewer 2 Report

The manuscript was modified according to the recommendations. It is fine and I do accept it in the present form.